# Influence of Layer Thickness and Extrusion Temperature on the Mechanical Behavior of PLA–Flax TPMS Sandwich Structures Fabricated via Fused Filament Fabrication

**DOI:** 10.3390/ma18235356

**Published:** 2025-11-27

**Authors:** Gabriele Marabello, Mohamed Chairi, Mariasofia Parisi, Guido Di Bella

**Affiliations:** 1Department of Engineering, University of Messina, Contrada di Dio, 98166 Messina, Italy; gabriele.marabello@studenti.unime.it (G.M.); mohamedchairi@cnr.it (M.C.); mariasofia.parisi@unime.it (M.P.); 2CNR ITAE, Salita S. Lucia sopra Contesse 5, 98126 Messina, Italy

**Keywords:** triply periodic minimal surface (TPMS), fused filament fabrication (FFF), PLA–flax composites, sandwich structures, additive manufacturing, bio-based materials, sustainability

## Abstract

**Highlights:**

**What are the main findings?**
Gyroid PLA–flax TPMS sandwiches printed via FFF were characterized in terms of flexure and compression. Flexural behavior was skin-dominated, with best strength at 0.28 mm/200 °C, and ANOVA confirmed the significant effects of temperature and layer height. Compressive behavior was core-dominated, showing cellular collapse with first-collapse stresses of 6.3–8.2 MPa, significantly governed by temperature and layer height (ANOVA).Layer height strongly influenced printing time/energy/CO_2_, with 0.28 mm providing the best kWh·MPa^−1^ efficiency.

**What are the implications of the main findings?**
Identifies process windows for optimizing TPMS sandwiches under multi-axial loading (skin vs. core-dominated regimes).Supports sustainable, energy-efficient manufacturing of bio-based lightweight components.

**Abstract:**

Triply periodic minimal surface (TPMS) sandwich structures made from PLA, reinforced with flax fibers, offer a bio-based approach to lightweight design, but their performance is sensitive to material-extrusion parameters. This study investigates the combined effects of layer height (0.16, 0.24, and 0.28 mm) and extrusion temperature (200, 220 °C) on the flexural behavior of gyroid-core PLA–flax sandwiches. Six parameter combinations were fabricated by fused filament fabrication and tested in three-point bending to obtain flexural strength and strain at failure. Post-fracture optical microscopy related mesostructure and failure mechanisms to macroscopic response. The highest strength (≈23 MPa) was found at 0.28 mm/200 °C, while the greatest strain at failure (≈0.06 mm/mm) occurred at 0.16 mm/200 °C. Two-factor ANOVA showed the significant main and interaction effects of temperature and layer height on both metrics. Fractography revealed a transition from interfacial delamination at lower temperatures and thinner layers to a more localized, cohesive rupture as interlayer bonding improved with higher temperature and thicker layers. Complementary compression tests revealed a core-dominated cellular collapse, with first-collapse stresses ranging from 6.3 to 8.2 MPa and a significant dependence on layer height and temperature (ANOVA). A gate-to-gate sustainability assessment indicated that layer height dominates printing time, energy demand, and CO_2_ emissions, with 0.28 mm minimizing energy per unit property. Measured part masses were 4–6% below slicer predictions, consistent with typical FFF porosity. The results provide TPMS-specific process windows that balance mechanical performance and energy efficiency for PLA–flax sandwiches.

## 1. Introduction

Triply periodic minimal surface (TPMS) lattices have emerged as architected materials that combine high specific stiffness/strength, tunable permeability, and smooth, curvature-continuous surfaces that reduce stress concentrations. Their sheet- and skeletal-based morphologies (e.g., Gyroid, Diamond, I-WP, Primitive) enable broad application spaces, from impact and blast mitigation and crashworthiness to acoustics and thermal management, while supporting functionally graded designs that tailor local response without geometric discontinuities [1,2,3,4,5,6]. Recent reviews further consolidate TPMS’ role as a unifying design class for lightweight, multifunctional structures across mechanical, civil and biomedical engineering, underscoring their geometrical controllability and multi-physics potential [7,8].

Additive Manufacturing (AM) is the key enabler that transforms these mathematically defined surfaces into physical parts. By exploiting “complexity for (nearly) free,” AM processes produce smooth, interconnected channels and thickened sheets that are otherwise impractical with subtractive or formative routes. Beyond simply “printing lattices”, AM allows the development of gradient architectures, topology-optimized features, and multiphysics-driven layouts that link relative density, energy absorption, and flow to local performance targets. This has catalyzed a wave of TPMS research using polymeric and metallic platforms and established robust evidence that AM parameters and design decisions co-govern mechanical response and reliability [9,10,11].

Coupling TPMS architectures with bio-based composites offers a promising approach to sustainable, high-performance components. PLA and natural fiber reinforcements (flax, hemp, kenaf, wood flour or particles, agricultural residues) offer favorable stiffness-to-weight ratios and a lower environmental footprint than purely petro-based systems, while additive manufacturing facilitates material efficiency and end-of-life strategies. Recent studies show that natural fibers can be processed as short, discontinuous fills in filaments, or, increasingly, embedded as continuous reinforcements. The TPMS geometry distributes loads, enhances failure tolerance, and enables tailored stiffness while preserving mass transport through the core [12,13].

Within the broader landscape of lightweight and bio-inspired cellular architectures, the Gyroid TPMS is part of a continuum that includes conventional honeycomb cores, strut-based lattices, and nature-inspired porous systems. Bioinspired lightweighting strategies derived from trabecular bone, beetle forewings, mantis shrimp exoskeletons and other biological templates have led to a wide range of architected materials that exploit hierarchical porosity, stiffness gradients, and tortuous load paths to achieve high specific stiffness and energy absorption [14,15].

Recent studies on 3D-printed bioinspired lattices in PLA demonstrate that these architectures can be tailored for improved compressive strength and impact energy absorption by appropriately tuning their internal topology [16].

Compared with traditional honeycomb cores, Gyroid and other TPMS lattices offer a more isotropic mechanical response and smoother stress distribution due to their curvature-based, node-free geometry, which reduces stress concentrations and enhances multi-axial load-bearing capability [17,18,19].

This positions Gyroid TPMS structures as promising alternatives to classical honeycomb and truss-like lattices in lightweight sandwich designs, particularly when complex loading conditions, multifunctionality or bio-based materials are considered.

Recent work on beetle elytron-inspired sandwich plates has shown that trabecular–honeycomb hybrid cores can achieve higher specific load-bearing capacity and specific energy absorption than cylindrical honeycombs with comparable wall thickness, further emphasizing the potential of bio-inspired cores in lightweight design [20].

However, realizing the full potential of TPMS–bio composite combinations requires careful control of manufacturing parameters that govern interfacial quality and mesostructural integrity. In fused-filament fabrication (FFF), nozzle and bed temperatures, raster orientation, layer height, printing speed, and infill strategy determine interlayer bonding, porosity, and fiber/matrix adhesion, factors that are especially sensitive when natural fibers are hydrophilic and thermally delicate. Even post-print hygrometry can degrade stiffness and shear properties or promote interlaminar fracture, emphasizing the need for parameter windows that enhance fiber bridging and matrix wetting while limiting moisture uptake and thermal damage [21,22,23,24].

A growing state-of-the-art focus on process optimization for bio-based FFF composites, including PLA-flax and other natural fibers. Taguchi and RSM-based designs, along with systematic parametric studies, have quantified how raster angle, infill density, temperature, and speed affect tensile, flexural, fatigue, and energy-absorption metrics. Complementary work explores hybrid and residual-biomass fills (e.g., husks, shells) to balance stiffness, damping, and sustainability. Collectively, these studies provide actionable guidance but also reveal material- and geometry-dependent trade-offs that must be re-examined for architected TPMS cores and sandwich configurations [25,26,27,28].

Calabrese et al. [29] studied FFF-printed gyroid-core sandwich structures made of PLA and PLA–flax, systematically varying deposition temperature and gyroid infill to link processing with flexural performance and fracture features. They found that tuning temperature and core density measurably improves skin–core adhesion and bending strength, providing process guidelines for TPMS + PLA/flax systems. Ayrilmis et al. [30] investigated wood-fiber/PLA green biocomposite panels featuring a gyroid TPMS core, focusing on how the face/core layer ratio affects bending, compression, hardness, and screw withdrawal resistance. They found that increasing the facesheet thickness significantly enhances mechanical properties and shifts failure modes, underscoring the role of skin design in TPMS-based bio-composites. Sharma and Le Ferrand [31] developed wood-PLA gyroid scaffolds that host mycelium growth, yielding strong, thermally insulating bio-composites for sustainable construction. They found that scaffold porosity (~90%) optimizes mycelium integration and multifunctional performance, illustrating how TPMS + natural-fiber-reinforced PLA architectures enable structural–thermal synergy.

In parallel, bio-inspired thin-walled tubes for building applications have recently been studied under axial and oblique impact loading, with crashworthiness metrics such as energy absorption and specific energy absorption optimized using multi-criteria decision-making (MCDM) approaches [32].

While those contributions primarily address dynamic crash scenarios, the present work focuses on quasi-static three-point bending as a dominant loading mode for sandwich beams, providing a complementary perspective that can later be integrated into broader performance and sustainability optimization frameworks.

This work investigates the influence of FFF process parameters on the mechanical response of PLA–flax TPMS gyroid structures, which serve as models for sandwich cores. Six printing configurations are created by combining two nozzle temperatures (200 and 220 °C) with three-layer heights (0.16, 0.24, and 0.28 mm) and evaluated in three-point bending for flexural strength and strain at failure. The highest flexural strength (≈23 MPa) is obtained at 0.28 mm/200 °C, whereas the lowest values (≈19 MPa) occur at 0.16 mm/200 °C and 0.24 mm/220 °C. The maximum strain at failure (≈0.06 mm/mm) is observed at 0.16 mm/200 °C. Optical microscopy of fractured specimens is employed to link mesostructure and failure mechanisms to the macroscopic response, with particular attention to interlayer decohesion, fiber pull-out, and local wall buckling across temperature–layer height combinations. An analysis of variance (ANOVA) is conducted on flexural strength and strain at failure to assess the statistical significance of the chosen factors and their interaction. To complement the flexural investigation, quasi-static compression tests are performed to capture the core-dominated behavior of the gyroid architecture, quantify the first-collapse stress, and assess the sensitivity of compressive response to layer height and extrusion temperature. ANOVA is likewise applied to compressive metrics to evaluate the main effects and the absence of significant interaction. A preliminary sustainability assessment, considering material usage, build time, energy demand, and waste, maps the printing configurations against performance, indicating trade-offs between process efficiency and mechanical properties. Overall, the study integrates mechanical testing, fracture microscopy, statistical design and analysis, and sustainability considerations to provide TPMS-specific temperature–layer height process windows for bio-based PLA–flax architectures fabricated via FFF.

This investigation complements and extends previous work by the authors on gyroid PLA–flax sandwiches, where core density was varied at fixed layer height, by now focusing on the combined influence of layer height and extrusion temperature at constant density. The new study includes a detailed fracture analysis and a sustainability assessment not covered in earlier research, thus constituting an independent and original contribution. In addition, the inclusion of compressive characterization provides a broader understanding of the multi-axial behavior of PLA–flax TPMS cores, highlighting the distinct process–structure relationships governing skin-dominated flexure and core-dominated collapse.

## 2. Materials and Methods

### 2.1. Sandwich Structure Design

Sandwich panels with a TPMS Gyroid core and continuous facesheets were additively manufactured for three-point bending tests using PLA–flax composite filament (Starflax 3D, Nanovia, Louargat, France; Ø 1.75 mm).

The PLA–flax composite filament (batch 08/2023) is a bio-sourced and biodegradable material containing approximately 10 wt% short flax fibers (<250 µm length, ≈15–25 µm diameter) uniformly dispersed in a PLA matrix. The material has a density of 1.25 g cm^−3^, a tensile modulus of 2.8–3.1 GPa, and a tensile strength of 37–41 MPa according to ISO 527-2/1A tests [33]. These data are consistent with the manufacturer’s technical sheet and with independent mechanical characterizations reported for similar Nanovia-grade PLA–flax filaments [26,27]. Microscopic examination of the fractured surfaces in this study confirmed the presence of short, discontinuous fibers embedded in the polymer matrix with observed lengths typically below 200 µm, consistent with the nominal range declared by the producer.

The gyroid geometry was generated in nTopology (nTop) via parametric modeling, allowing explicit control of unit cell size and wall thickness to meet the target specimen envelope for flexural testing. The core relative density was fixed at 30% in CAD by setting the gyroid wall thickness, in accordance with the results of [29].

The gyroid is a triply periodic minimal surface (TPMS) with zero mean curvature, that provides a smooth, intersection-free architecture and a favorable strength-to-weight ratio. A common trigonometric approximation is:sin(x)cos(y) + sin(y)cos(z) + sin(z)cos(x) = 0.(1)

Models were exported as STL files (triangulated meshes with vertex coordinates and outward normals) and processed in the slicer to generate G-code for fabrication.

During slicing, walls were set to 100% infill, meaning the internal regions of the gyroid walls were rendered fully solid. This preserves the designed wall thickness specified in CAD and ensures the printed core accurately reproduces the intended section, minimizing internal voids and maintaining structural integrity and mechanical performance.

After slicing, the models were transferred to the printer for fabrication. Each specimen was oriented with its smallest dimension along the Z-axis (i.e., horizontal layers at 90° to the bending load). No post-processing (chemical, thermal, or finishing) was applied.

An as-printed PLA–flax TPMS gyroid sandwich specimen (30% relative density) is shown in Figure 1.

### 2.2. Additive Manufacturing of Test Specimens

Test specimens were produced by Fused Filament Fabrication on a Creality K1 Max (stock hardware, 0.4 mm nozzle) (Shenzhen Creality 3D Technology Co., Ltd., Shenzhen, China). Slicing and toolpath generation were performed in Bambu Studio v. 2.3.1.51 and then exported to G-code. Print jobs were batched, with three specimens fabricated simultaneously per build to equalize thermal history and airflow conditions; batches were repeated until the target sample size was reached.

All printing parameters except extrusion temperature and layer height were kept constant to ensure process repeatability. The samples were produced using a rectilinear infill pattern for the facesheets (100% infill). The build plate temperature was maintained at 60 °C, with the cooling fan set to 100% after the first layer. The main fixed settings included a first-layer speed of 50 mm/s, outer-wall speed of 200 mm/s, inner-wall and infill speeds of 270–300 mm/s, and top-surface speed of 200 mm/s. Three wall loops were used (≈1.2 mm thickness) with 0.8 mm retraction at 30 mm/s and no build-plate adhesion structures. These parameters were verified for all print runs to ensure consistent extrusion conditions and to isolate the effects of temperature and layer height on the final mechanical and microstructural response.

The printing window comprised two extrusion temperatures (200 and 220 °C) and three layer heights (0.16, 0.24, and 0.28 mm); the build plate was maintained at 60 °C. Unless otherwise specified, line widths followed the nozzle diameter, and perimeter and skin counts were kept constant to isolate the effects of temperature and layer height. Flow tuning, first-layer calibration, retraction (direct-drive), fan management, and acceleration limits were fixed after an initial calibration series and held unchanged for all subsequent batches.

Filament was conditioned in a dry box at 50 °C for 6 h before and between print runs to remove residual moisture and ensure consistent extrusion. This pre-drying step, commonly adopted for PLA-based filaments, prevents humidity-induced porosity and improves interlayer adhesion. Similar conditioning parameters have been reported to effectively stabilize the moisture content of PLA prior to additive manufacturing [34]. The build surface (glass/PEI) was cleaned with isopropyl alcohol prior to each job; no rafts or supports were used, and a short brim was only added when first-layer adhesion checks indicated this was needed. After printing, parts cooled on the plate to limit warpage; no chemical, thermal, or surface finishing was applied. Dimensional checks on the first part of each batch targeted ±0.2 mm for thickness and width, and batch-to-batch mass dispersion was kept below 2%. Each specimen was labeled PF-T-H, where T denotes the extrusion temperature (200 or 220 °C) and H the layer height in hundredths of a millimeter (e.g., PF-220-24).

To ensure repeatability, the full parameter set is reported in Table 1.

### 2.3. Mechanical Testing

Flexural characterization was conducted according to ASTM D790 [35]. Tests were performed on a ZwickRoell universal testing machine (Ulm, Germany) equipped with a 2.5 kN load cell, using a support span of 120 mm and nominal specimen dimensions of 150 × 40 × 12 mm. Displacement was applied at 0.1 mm/min until failure. For statistical robustness, 18 specimens (2 extrusion temperatures × 3 layer heights × 3 repeats) were tested. Stress and strain values were calculated according to the standard using the following equations:(2)$$\sigma =\frac{3FL}{2bd^{2}}$$(3)$$\varepsilon =\frac{6Dd}{L^{2}}$$
where F is the applied load, L is the span length, b is the specimen width, d is its thickness, and D is the deflection at mid-span.

Fractographic inspection and failure-mode assessment were conducted with a Hirox KH-8700 digital microscope (Tokyo, Japan).

To assess the influence of process parameters on flexural response, an ANOVA was performed, with α = 0.05. When significant effects were detected, Tukey’s HSD post hoc comparisons were used to identify pairwise differences.

A complementary sustainability evaluation was carried out by combining (i) material consumption exported from the slicer (part mass and waste), (ii) printing time and energy use estimated from the printer’s power profile and job logs, and (iii) the measured mechanical performance. These indicators were used to discuss process–property–impact trade-offs for the 30%-density TPMS sandwich specimens.

Compression characterization was performed according to ASTM C365 [36] to evaluate the core-dominated response of the gyroid architecture. Square specimens (75 × 75 × 12 mm) were tested using the same ZwickRoell universal testing machine and 2.5 kN load cell employed for flexure. Displacement was applied at a constant rate of 2 mm/min until densification. For each of the six printing configurations, three specimens were tested (*n* = 18). Compressive stress and strain were computed following the standard using:(4)$$\sigma =\frac{P}{A}$$(5)$$\varepsilon =\frac{d}{h_{0}}$$
where P is the applied load, A the loaded area (75 × 75 mm^2^), d the displacement, and h_0_ the initial specimen thickness.

Failure morphology after the first collapse and at densification was documented with the Hirox KH-8700 digital microscope (Hirox Co., Ltd., Akishima, Japan). To quantify the influence of process parameters on compressive performance, a two-factor ANOVA (α = 0.05) was performed on the first-collapse stress.

## 3. Results

### 3.1. Preliminary Analysis

Table 2 summarizes the slicer outputs for each configuration, where three specimens are always printed simultaneously under combinations of layer heights of 0.16, 0.24, and 0.28 mm, and nozzle temperatures of 200 or 220 °C. As expected, build time decreases with increasing layer height (from 8 h 45 min at 0.16 mm to 5 h 56 min at 0.28 mm, a reduction of ≈32%), whereas the theoretical mass of the three parts (Wt) and the filament length remain almost unchanged with nozzle temperature, since the slicer estimation depends solely on geometry and deposition path. Small variations in Wt between layer heights (≈113–119 g per batch) are related to path-planning adjustments (line width, top/bottom compensation, and rounding of thin features) that slightly modify the number of extruded roads at constant CAD volume [37,38,39].

Table 3 compares the measured masses with slicer predictions. For each configuration, the individual specimen masses are virtually identical (St. Dev. = 0), confirming highly repeatable dosing during parallel printing. However, the total real mass (Wr) is consistently lower than Wt by about 4–6% (ΔW = −4.2 to −5.7%). Two main process trends can be observed:Effect of layer height. The largest mass deficit appears at 0.24 mm (≈−5.7% at 200 °C), whereas 0.16 mm and 0.28 mm yield smaller discrepancies (−4.2 to −4.4%). This is consistent with different road-packing efficiencies, where changes in raster gap and overlap determine the inherent porosity and thus lower the effective part density [37]. This indicates that porosity is slightly more pronounced at the intermediate layer height, where the bead geometry and packing efficiency lead to less effective filling of the TPMS cavities.Effect of temperature. Increasing the nozzle temperature from 200 to 220 °C does not affect Wt (by definition) and only marginally influences Wr: porosity reduction requires, not only higher temperature but also flow-rate or overlap correction; otherwise, void topology remains dominated by the road geometry and nominal flow [38].

**Table 3 materials-18-05356-t003:** Real weight Wr vs. theoretical Wt.

Configuration	Mean Weight [g]	St. Dev. [g]	Wr [g]	Wt [g]	ΔW [%]
200 °C|0.16 mm	36.6	0.00	109.8	114.61	−4.2
200 °C|0.24 mm	35.6	0.00	106.8	113.25	−5.7
200 °C|0.28 mm	37.8	0.00	113.4	118.64	−4.4
220 °C|0.16 mm	36.6	0.00	109.8	114.61	−4.2
220 °C|0.24 mm	36.1	0.00	108.2	113.25	−4.4
220 °C|0.28 mm	37.9	0.00	113.6	118.64	−4.2

Continuing from these trends, the systematic shortfall of Wr relative to Wt can be explained by how slicers estimate mass (idealized, fully dense roads at nominal flow) versus how FFF actually deposits material. In practice, the mesostructure contains persistent inter- and intra-layer voids governed by raster gap/overlap, start/stop transients, and local path geometry; as a result, the apparent density of “solid” regions is typically a few percent below the bulk filament density [37,40,41]. Moreover, small deviations in dosing (i.e., a slightly conservative flow or narrower realized line width) further reduce printed volume at constant envelope; adjusting the extrusion multiplier or overlap is known to recover mass and density and diminish porosity, in agreement with controlled studies that tune flow rate and line width [40,42]. Parameter screens that explicitly include flow rate alongside layer height, speed, and temperature also report significant changes in porosity and packing efficiency of the roads, reinforcing that flow control rather than temperature alone is the primary lever for densification [41]. Finally, for PLA–flax and other bio-based filaments, micro-porosity within the extrudate and imperfect fiber wet-out can depress the effective density even when the external geometry is accurate, which is consistent with the modest 4–6% mass deficit measured here [43]. A moderate increase in nozzle temperature can aid interfacial fusion, but without concomitant flow or overlap adjustments, it does not guarantee a measurable rise in bulk density, again matching the weak temperature sensitivity observed across configurations [44].

From a mechanical standpoint, the 4–6% deficit between Wr and Wt implies a non-negligible but relatively uniform level of porosity across all settings, mainly governed by the gyroid architecture and the material-extrusion process. Within this narrow range, the configuration with the highest mass deficit (0.24 mm, 200 °C) also exhibits one of the lowest flexural strength and stiffness values, consistent with a slightly lower effective density. However, configurations with similar mass deficits (e.g., 0.16 mm and 0.28 mm) display markedly different flexural responses because layer height modifies the number of interfaces, the skin/core morphology, and the failure mode under bending. As a result, small differences in mass deficit should be considered a coarse indicator of porosity rather than a primary design parameter, while geometric and interfacial effects associated with layer height and fracture mechanisms remain the dominant factors controlling flexural behavior.

### 3.2. Three Points Flexural Tests

The flexural response of the TPMS-sandwich specimens is examined through representative stress–strain curves obtained under three-point bending. The analysis focuses on the characteristic regions of the curves and on how processing variables affect stiffness, peak stress, and post-peak evolution. To this end, Figure 2a reports the family of curves at 200 °C for different layer heights, while Figure 2b presents the corresponding set at 220 °C. The two figures are discussed comparatively to isolate the role of layer height at a fixed temperature and the role of temperature at a fixed geometry, enabling clear attribution of trends to process conditions. Quantitative observations are drawn directly from the plotted data, and qualitative interpretation is limited to the shape and progression of the curves.

The representative three-point-bending response of these TPMS sandwiches shows: (i) an initial linear region where the slope (apparent flexural modulus) is nearly constant; (ii) a nonlinear rise to a distinct maximum (peak stress), where the tangent stiffness drops to approximately 0; and (iii) a post-peak segment where the load-carrying capacity decreases with strain. This staging is consistent with standard ASTM D790 characterization and serves as the template for interpreting all subsequent curves in this study [29]. This common set of descriptors enables a consistent, parameter-by-parameter comparison and prevents over-interpretation of local fluctuations in the raw signal. With this framework in place, the family of curves at 200 °C can be examined to quantify how varying only the layer height (0.16, 0.24, and 0.28 mm) modulates these metrics under otherwise identical conditions (see Figure 2a).

All three curves closely overlay at very small strain, indicating comparable global stiffness in the purely elastic segment. Slight differences emerge as strain increases: the 0.28 mm condition tends to maintain a higher instantaneous slope for longer before deviating from linearity, whereas the 0.16 mm and 0.24 mm conditions depart toward nonlinearity earlier. This suggests a modest stiffening benefit at equal temperature when the layer height is increased. Similar sensitivities of elastic and early-plastic responses to printable bead geometry and inter-bead continuity have been documented for PLA under bending [44,45].

At 200 °C, the maximum stress increases with layer height: 0.16 mm and 0.24 mm reach similar peaks (~19–20 MPa), while 0.28 mm peaks higher (about 23–24 MPa). The onset of nonlinearity occurs at comparable strain for all three batches, but the 0.28 mm trace sustains a larger stress increment before reaching the maximum. This “higher-peak at larger layer height” trend aligns with studies identifying layer height as a primary factor for flexural strength in FFF PLA, due to its influence on road cross-section, road overlap, and the effective load-bearing area of the outer skins [45,46].

Beyond the maximum, the softening slope steepens progressively from 0.16 to 0.24 to 0.28 mm. The 0.16 mm condition exhibits the longest tail (greater strain capacity after peak), while 0.28 mm shows the sharpest drop (shorter usable ductility). In practical terms, increasing layer height at 200 °C trades some post-peak tolerance for higher peak stress. This trade-off is coherent with broader MEX/FFF literature where coarser vertical discretization reduces the number of interlayer planes, alters neck geometry between adjacent rasters, and generally shifts the balance toward higher maxima with less gradual post-peak decay [44,45].

Well-established processing–structure relationships explain the observed family of curves at 200 °C.

Larger layer height reduces the number of layer interfaces across the thickness and increases the cross-section of each deposited road. Under bending, this can increase the effective load-bearing area of the outer skins and shift the peak upward, consistent with the higher maxima at 0.28 mm. Multiple investigations rank layer height among the most influential parameters for flexural strength in PLA parts, often ahead of raster angle when other settings are held constant [46].

The softening portion of the curve is highly sensitive to the continuity and morphology of adjacent filaments (overlap ratio, local necking, and bead geometry), which are themselves governed by layer height and temperature. Studies correlating temperature field, interlayer coalescence, and elastic-modulus trends in PLA confirm that processing conditions which enhancing filament fusion tend to increase maxima and alter the subsequent decay of the stress–strain trace [44,47,48].

While the sandwich architecture here is sheet-based TPMS, the macroscopic stress–strain shape under bending (linear → peak → softening) remains consistent with prior gyroid-focused studies combining experiments and simulations. Those works emphasize that, at a fixed core density, process parameters that affect the skins and skin–core continuity dominate the global bending curve metrics, consistent with the trends observed at 200 °C [5].

Similarly, the family of curves at 220 °C can be examined (see Figure 2b).

In the small-strain segment, the three traces essentially overlap, indicating comparable apparent flexural stiffness at 220 °C across layer heights. The departure from linearity occurs at nearly the same strain for all batches (≈0.025–0.03 mm/mm), suggesting that at this temperature, early-stage stiffness is weakly sensitive to vertical discretization and is mainly governed by the consolidated skins and the gyroid core acting as a shear spacer. The modest tendency of the 0.16 mm and 0.28 mm curves to maintain a higher instantaneous slope up to the onset of nonlinearity is observable and is consistent with temperature-assisted interroad coalescence improving effective skin continuity at 220 °C. Such temperature effects on elastic response and bonding quality in FFF PLA are documented by process-structure studies that track the temperature field and its impact on neck growth and modulus [48].

The maximum stress shows clear stratification: the 0.28 mm condition reaches the highest peak (≈22–23 MPa), the 0.16 mm curve follows closely (≈21–22 MPa), and the 0.24 mm trace exhibits a distinctly lower maximum (≈18–19 MPa). Strain at peak clusters around ≈0.038–0.042 mm/mm for all three batches. The strength ranking aligns with the role of layer height as a primary factor for flexural strength in material-extrusion PLA, via its effect on road cross-section and the effective load-bearing area of the skins; elevated nozzle temperature further promotes interlayer diffusion and coalescence, which can sustain higher peak stresses when geometric conditions are favorable [44,45].

Beyond the maximum, all curves display a relatively short post-peak segment at 220 °C. The 0.24 mm case shows the steepest drop (shortest usable deformation after peak), while 0.16 and 0.28 mm retain slightly longer but still limited tails. The compactness of the softening region at this temperature indicates that, once the maximum is reached, residual load-carrying capacity decays rapidly for all layer heights. The sensitivity of the softening behavior to processing temperature, through its influence on filament wetting, interfacial consolidation, and the continuity of adjacent roads, is consistent with controlled studies that connect higher nozzle temperatures with improved maxima yet sharper post-peak evolution depending on the printed morphology [44,48].

Material-extrusion studies consistently report that (i) increasing layer height can raise flexural strength by enlarging the road cross-section and reducing the number of interlayer planes across the thickness, and (ii) higher extrusion temperature lowers melt viscosity, enhances interlayer diffusion, and strengthens interfacial bonding-factors that influence both the location of the peak and the subsequent decay of the curve. These mechanisms underpin the observed family of curves at 220 °C and align with parametric investigations on PLA and architected-core systems where skins dominate the macroscopic bending response [5,45].

Table 4 presents, in a compact form, the key features extracted from the stress–strain families in Figure 2a,b, using a common set of descriptors (peak stress and qualitative post-peak character) for direct cross-comparison.

Taken together, the entries show that at 200 °C the response scales monotonically with layer height—0.28 mm delivers the highest peak with the shortest post-peak segment, 0.16 mm the lowest peak with the most extended tail, and 0.24 mm remains intermediate—whereas at 220 °C the ranking changes subtly: 0.28 mm retains the highest peak (≈22–23 MPa) and a short post-peak, 0.16 mm follows closely with a similarly compact softening, and 0.24 mm exhibits both a lower maximum and the steepest decay. Comparing temperatures at fixed geometry, raising T from 200 °C to 220 °C tends to compress the post-peak region for all cases, increases the peak for 0.16 mm, leaves 0.28 mm broadly high but slightly reduced, and penalizes 0.24 mm. Operationally, the coarse discretization (0.28 mm) is the most effective for maximizing peak stress at both temperatures, the fine discretization (0.16 mm) offers greater post-peak deformation capacity at 200 °C but converges to a short tail at 220 °C, and the mid-height setting (0.24 mm) emerges as the least robust when the temperature is elevated.

Figure 3a summarizes the maximum flexural stresses for each layer height at both temperatures, with error bars showing one standard deviation. At 200 °C, the ranking is monotonic with layer height: 0.28 mm = 23.2 ± 0.21 MPa (highest), while 0.16 mm = 19.3 ± 0.47 MPa and 0.24 mm = 19.3 ± 0.25 MPa are coincident within the scatter. At 220 °C, the pattern becomes bimodal: 0.16 mm = 22.2 ± 0.61 MPa and 0.28 mm = 22.2 ± 0.36 MPa share the highest value, whereas 0.24 mm = 19.0 ± 0.29 MPa remains distinctly lower. Comparing temperatures at fixed geometry, increasing T from 200 to 220 °C raises the peak for 0.16 mm by +2.9 MPa (+15.0%), leaves 0.24 mm essentially unchanged (−0.3 MPa; −1.6%), and slightly reduces 0.28 mm (−1.0 MPa; −4.3%).

It is important to note that the slight reduction in flexural strength observed at 220 °C is not related to any thermal degradation of the flax fibers. Thermogravimetric analysis (TGA) of the same PLA–flax filament, as reported by Calabrese et al. [29], showed no measurable mass loss below 230–240 °C, confirming the thermal stability of both the PLA matrix and the flax reinforcement within the investigated temperature range. Therefore, the performance variation at 220 °C is attributed to process-related factors such as local over-melting, reduced dimensional stability, and minor geometric distortion of the TPMS walls rather than to fiber degradation.

Dispersion is uniformly small (coefficients of variation: 0.9–2.7%), supporting the robustness of these trends. Operationally, coarse discretization (0.28 mm) maximizes strength at 200 °C and remains high at 220 °C; fine discretization (0.16 mm) benefits markedly from the higher temperature; the intermediate setting (0.24 mm) underperforms at both temperatures.

Figure 3b reports the strain at maximum load for all conditions, with error bars representing one standard deviation. At 200 °C, the mean strain decreases from 0.060 ± 0.006 (0.16 mm) to 0.050 ± 0.007 (0.24 mm) and 0.050 ± 0.0054 (0.28 mm), indicating that coarser vertical discretization shortens the deformation capacity at the peak.

At 220 °C, the values more tightly clustered—0.040 ± 0.0007 (0.16 mm), 0.050 ± 0.005 (0.24 mm), and 0.040 ± 0.0008 (0.28 mm)—so the 0.16 mm and 0.28 mm settings converge to approximately 0.04, while 0.24 mm remains around 0.05. Comparing temperatures at fixed geometry, increasing T from 200 to 220 °C reduces the peak strain significantly for 0.16 mm (−0.020, −33%) and 0.28 mm (−0.010, −20%), whereas 0.24 mm is essentially unchanged (Δ ≈ 0.00). Variability analysis confirms the trend: coefficients of variation at 200 °C are about 10% (0.16 mm), 14% (0.24 mm), and 11% (0.28 mm), while at 220 °C they drop to 1.8–2.0% for 0.16 mm and 0.28 mm and remain, about 10% for 0.24 mm. In practical terms, the higher temperature compresses the strain-at-peak for the fine and coarse layer heights, yielding tighter, more repeatable maxima, whereas the intermediate 0.24 mm condition preserves a higher peak strain but with comparatively larger scatter.

To assess whether temperature (T), layer height (h), and their interaction (T × h) have statistically significant effects on the flexural metrics, a two-way ANOVA is considered. The response variables are the maximum stress and the strain at maximum load; factors are fixed with two levels for T (200, 220 °C) and three levels for h (0.16, 0.24, 0.28 mm). The analysis follows the usual workflow: verification of model assumptions (approximate normality of residuals and homoscedasticity), computation of F-ratios for the two main effects and the interaction at α = 0.05, and reporting of effect sizes (e.g., η^2^ or partial η^2^). The forthcoming ANOVA plots (means with confidence intervals, interaction plots, residual diagnostics) are interpreted within this framework to determine which process variable(s) drive the observed differences and how robust those differences are.

The residual diagnostics for stress are satisfactory (see Figure 4). The normal probability plot is close to linear with only mild deviations at the extremes, suggesting at most a slight light-tailed behavior; no pronounced outliers are evident. The histogram is roughly symmetric and centered near zero, again consistent with approximate normality. In the residuals vs. fits panel there is no curvature or funneling—residual spread remains essentially constant from nearly 19 to 23 MPa—so heteroscedasticity is not indicated; the visible vertical clusters simply reflect the discrete treatment means. The residuals vs. order plot shows an alternating, patternless scatter around zero with no drift over time; the last positive residual (about +0.6) is isolated but not influential. Overall, the ANOVA assumptions of normality, homoscedasticity, and independence appear reasonable, and no transformation is warranted for the stress response.

The ANOVA for stress, reported in Table 5, indicates excellent model fit (R^2^ = 96.66%, R^2^(adj) = 95.26%, R^2^(pred) = 92.48%; residual SD S = 0.390 MPa). Both temperature (T) and layer height (h) are statistically significant, as is their interaction T × h. Layer height is the dominant contributor (F = 124.75, *p* < 0.001; Adj SS = 38.002 MPa^2^, ≈69.5% of total variance), while temperature has a smaller but still significant effect (F = 8.24, *p* = 0.014; Adj SS = 1.255 MPa^2^, ≈2.3%). The interaction is large and highly significant (F = 44.59, *p* < 0.001; Adj SS = 13.583 MPa^2^, ≈24.8%), confirming that the influence of T depends on the chosen layer height and vice versa. The small error term (Adj MS = 0.152 MPa^2^) relative to the treatment mean squares supports good repeatability and the robustness of the detected effects. In practical terms, h drives most of the strength variation, but because T × h is sizable, strength rankings change with temperature (e.g., the advantage of coarse layers at 200 °C and the bimodal 0.16/0.28 top values at 220 °C), so factor settings must be selected jointly rather than optimized in isolation.

The significant T × h interaction can be explained by the thermal history and diffusion dynamics at the layer–layer interface. At the smallest layer height (0.16 mm), the greater number of interfaces per unit thickness and the shorter time between neighboring passes favor heat accumulation and delay cooling of the previously deposited filament. Under these conditions, increasing the extrusion temperature to 220 °C lowers the melt viscosity of the PLA–flax composite and increases chain mobility, enhancing interlayer wetting and polymer interdiffusion at the interface and reducing voids. The resulting improvement in interlaminar bonding leads directly to the marked increase in flexural strength observed for 0.16 mm at 220 °C. In contrast, for the thickest layers (0.28 mm), the larger bead cross-section and the longer thermal path to the part interior promote a steeper temperature drop between successive layers; the outer region of each filament solidifies more rapidly, and the interface temperature at the moment of deposition is closer to, or below, the optimal healing window. In this regime, the benefit of a higher nozzle temperature is limited by insufficient time above the glass-transition temperature, so polymer reptation across the interface is constrained and the strength gain is small.

This behavior is in line with process–structure studies on PLA-based material extrusion, which show that interlayer adhesion depends on the combined effects of nozzle temperature, layer thickness, and interlayer time through their impact on interface thermal history and diffusion-driven healing, rather than on nozzle temperature alone [44,48]. It is also consistent with recent analyses of temperature-sensitive parameters and interfacial bonding in FFF parts, which highlight diminishing returns when available diffusion time is limited, even at elevated extrusion temperatures [49,50].

The residual diagnostics for strain at maximum load are fully acceptable (see Figure 5). The normal probability plot is nearly linear with only minor deviations in the lower tail, indicating approximate normality and no outliers of concern.

The histogram is narrow and centered close to zero, consistent with a small residual variance. In the residuals vs. fits panel there is no curvature or funneling across the fitted range (~0.041–0.061), so homoscedasticity holds; the vertical groupings reflect the discrete treatment means rather than a modeling issue. The residuals vs. order plot shows an alternating scatter around zero without drift or cycles, supporting independence over the acquisition sequence. Overall, the assumptions for ANOVA (normality, constant variance, independence) appear well satisfied for the strain response; no transformation is warranted.

The ANOVA for strain at maximum stress, reported in Table 6, shows a moderate model fit (R^2^ = 76.39%, R^2^(adj) = 66.55%, R^2^(pred) = 46.87%; residual SD S = 0.00484). The dominant factor is temperature (F = 27.18, *p* < 0.001; Adj SS = 0.000636), accounting for ~53% of the total variability in strain. Layer height alone is not significant at α = 0.05 (F = 1.13, *p* = 0.355; Adj SS = 0.000053, ~4% of total), but the T × h interaction is significant (F = 4.69, *p* = 0.031; Adj SS = 0.000219, ~18% of total), indicating that the effect of layer height depends on temperature. This is consistent with the sample means: at 200 °C strain decreases from 0.16 to 0.24/0.28, whereas at 220 °C, the 0.16 and 0.28 levels converge to approximately 0.04, while 0.24 remains around 0.05. The relatively lower R^2^(pred) suggests some unexplained dispersion (expected for a deformation metric measured at the peak), but the inference is clear: ductility at peak is governed primarily by temperature, with layer height exerting a secondary, temperature-dependent influence. Therefore, control of strain capacity should prioritize temperature setting, with layer height adjusted conditionally.

For the specimen printed at 200 °C with a 0.16 mm layer height, as observed in Figure 6, the fracture sequence initiates at the tensile skin (bottom layer) and propagates upward through the core, showing a typical bending-induced failure pattern.

At the early stage of loading, small surface defects and printing-induced voids in the outer skin act as stress concentrators. A primary crack nucleates at mid-span in the lower skin (central circled region) and propagates along the raster direction of the deposited filaments. As deformation increases, the crack progressively opens, causing local separation of the bottom skin and limited delamination at the core–skin interface, as evidenced by the secondary circled cracks on either side. The fracture then turns upward into the gyroid wall, following a tortuous path governed by stress redistribution within the core.

The magnified view reveals the detailed morphology of the fracture path: a sequence of brittle rupture zones alternating with ductile tearing segments, typical of FFF-printed PLA composites with partial interlayer fusion. The microstructural roughness and fibrillation observed inside the crack suggest that failure occurred mainly by inter-layer decohesion rather than by intra-layer rupture. This indicates a limited degree of polymer diffusion between successive layers at 200 °C, consistent with the lower deposition temperature leading to incomplete molecular interpenetration. The presence of small voids and unbonded regions at the filament boundaries further confirms that interfacial adhesion, both between deposited lines and between the core and the skin, plays a dominant role in crack propagation under flexural stress.

Overall, the observed mechanism corresponds to a progressive and non-catastrophic failure, characterized by sequential opening of the lower skin, partial delamination of the interface, and gradual advance of the fracture front toward the upper layers of the core, in line with the behavior described for the PLA–flax sandwich structures in the reference study [29].

For the 200 °C/0.24 mm configuration shown in Figure 7, the fracture pattern remains consistent with tensile-skin-initiated failure, but the damage is more extensive and structurally integrated with the core than in the 0.16 mm case.

The sequence begins with crack nucleation at mid-span in the lower skin, where tensile stresses are maximal. The fracture initiates along a single filament track and then spreads laterally following the raster orientation. The detachment front advances along the skin–core interface, creating a partial delamination zone, as indicated by the secondary circled cracks on both sides of the midline. As opening increases, the primary circled crack kinks upward into the first gyroid wall, where stress redistribution within the core drives a tortuous propagation path (see right-hand detail).

The two detailed views show complementary aspects of this mechanism.

On the left, the surface morphology displays a stepped pattern associated with interlayer decohesion, typical of limited polymer diffusion at 200 °C. The presence of smooth facets alternating with micro-bridged ligaments suggests a mixed adhesive–cohesive failure, where partial bonding resisted before final rupture.On the right, the crack tip exhibits evidence of progressive tearing and filament pull-out, indicating that the damage evolved gradually rather than catastrophically. This corresponds to the observed load–displacement behavior, in which stress decreases smoothly after the maximum load.

Compared with the 0.16 mm specimen, the 0.24 mm layer height leads to slightly larger delaminated regions and a more tortuous crack path through the interface, consistent with modestly improved interlayer fusion but still limited adhesion at this deposition temperature. The failure remains dominated by tensile rupture of the lower skin followed by localized delamination and upward crack propagation through the gyroid walls, confirming a progressive, non-brittle fracture mode typical of PLA sandwiches fabricated at 200 °C.

For the 200 °C/0.28 mm configuration, reported in Figure 8, the fracture mode maintains the general pattern observed at lower layer heights but reveals more cohesive and localized failure in the tensile skin, along with reduced interfacial delamination.

In the main view, a primary crack nucleates at mid-span in the lower skin (central circled region) and remains relatively confined. The fracture front then propagates upward through the gyroid wall along a single, well-defined path (see right-hand detail). The absence of extended delamination at the skin–core interface, apart from the small secondary circled cracks, indicates stronger interlayer bonding, likely due to the reduced number of interfaces per unit thickness and the higher thermal retention of the thicker extruded lines, which enhances filament–filament fusion.

The microscopic details (bottom images) provide further evidence of this enhanced bonding.

On the left, the lower skin fracture surface exhibits a rough, fibrous morphology, indicating cohesive failure within the extruded filaments rather than interlayer separation. The limited presence of planar decohesion planes contrasts with the more layered appearance observed in the 0.16 mm and 0.24 mm specimens.On the right, the propagation front within the core shows a narrow, continuous tear through successive gyroid walls, confirming that load transfer remained effective between the core and the skin up to the point of rupture.

The inset (top left) highlights a partial detachment occurring adjacent to the crack path, possibly related to local stress redistribution after skin rupture, but it does not evolve into extended delamination.

Overall, the fracture morphology of this sample supports a more integrated structural response, with higher load-bearing capability before failure and less progressive damage evolution than thinner-layer counterparts. The failure is still driven by tensile rupture of the bottom skin, but the increased layer thickness contributes to a more cohesive, stronger bond network, which delays crack propagation and explains the higher maximum load recorded for this configuration compared to 0.16 and 0.24 mm.

For the 220 °C/0.16 mm configuration, reported in Figure 9, the fracture exhibits a distinctly more cohesive and concentrated failure mode, consistent with the improved interlayer adhesion expected at the higher deposition temperature.

As shown, the crack initiates at mid-span in the lower skin (circled region), where bending tensile stresses are highest. The separation front is narrow and well confined, unlike the broader damage observed at 200 °C. After initiation, the primary crack kinks upward into the core (curved red arrow), following a straighter, more continuous path that indicates homogeneous stress transfer across the skin–core interface and limited interfacial delamination.

The magnified view confirms this behavior: the fracture front appears compact and sharply defined, with limited signs of interlayer decohesion. The tearing pattern is dominated by cohesive rupture of the extruded filaments, indicating effective molecular interdiffusion between adjacent layers due to the higher temperature. The smoother transition between the fractured skin and the adjacent gyroid walls reveals that debonding at the interface is minimal, and the damage evolves primarily through the core structure.

Overall, the fracture morphology at 220 °C and 0.16 mm indicates that increasing the temperature enhances interlayer bonding, reduces premature delamination, and results in a more localized and cleaner crack path. This behavior is consistent with a higher strain at failure and a more stable post-peak response, reflecting the improved adhesion between the skin and core and the stronger cohesive integrity of the printed structure.

For the 220 °C/0.24 mm configuration, reported in Figure 10, the fracture sequence maintains the typical pattern of tensile-skin initiation but shows a more complex and energy-dissipative crack evolution compared with the thinner-layer case, suggesting a balanced interaction between enhanced interlayer bonding and localized interfacial debonding.

Failure initiates at mid-span in the lower skin (primary circled crack). The microcrack follows the filament tracks and opens in a controlled manner. A shallow delamination develops at the skin–core interface, evidenced by the secondary circled cracks adjacent to the midline. The fracture then kinks upward into the first gyroid wall, remaining localized; the close-ups show limited interfacial separation with small bridging ligaments, consistent with improved interlayer bonding at 220 °C.

The magnified views provide further insight into the fracture mechanisms.

On the left, the rupture path exhibits alternating brittle and ductile features, with small unbroken filaments bridging the two fracture surfaces—evidence of partial filament coalescence and controlled energy dissipation.On the right, the cross-sectional morphology highlights the tight contact between skin and core, with limited interfacial voids and no extended detachment, confirming effective stress transfer during bending.

Overall, the fracture mode at 220 °C and 0.24 mm layer height reflects a hybrid mechanism in which tensile cracking of the skin remains the governing factor, but the higher deposition temperature enhances molecular diffusion and fusion between printed layers, resulting in a more cohesive fracture surface and a smoother post-peak load decay. The damage evolves in a progressive yet more localized manner, in line with the improved mechanical stability observed in the corresponding flexural response.

For the 220 °C/0.28 mm configuration, reported in Figure 11, the fracture exhibits the most cohesive and structurally integrated failure mode among all the tested conditions, highlighting the beneficial combined effect of higher deposition temperature and larger layer thickness on interlayer adhesion and overall structural integrity.

In the main view, a primary crack nucleates at mid-span in the lower skin (central circled region) and follows the filament tracks before kinking upward into the gyroid wall (curved red arrow). The crack path is compact and centrally confined, indicating a highly localized stress field. No extended delamination is visible at the skin–core interface; only small secondary circled cracks appear adjacent to the initiation site. This morphology confirms strong skin–core bonding at 220 °C and the cohesive nature of failure for the 0.28 mm layer.

The microscopic details provide additional evidence of this cohesive fracture mode:

On the left, the rupture surfaces display fused, plastically deformed filaments with limited interlayer separation, suggesting that failure occurred mainly by cohesive tearing rather than interfacial decohesion. The rough, fibrous texture reveals strong interdiffusion and thermal bonding between adjacent layers.

On the right, the crack front through the gyroid wall shows a continuous rupture path that cleanly connects the core and the skin, with no evidence of void formation or debonding. The core remains structurally intact except for the localized fracture zone, confirming its efficient load-transfer role during bending.

Overall, the 220 °C/0.28 mm sample demonstrates a highly cohesive and energy-absorbing fracture behavior, characterized by localized tensile rupture of the bottom skin followed by stable crack propagation through the gyroid wall. Enhanced thermal fusion between deposited filaments leads to a strong skin–core adhesion and suppresses interfacial delamination. This morphology is consistent with the highest maximum load recorded among the tested conditions, reflecting improved mechanical performance and failure stability associated with this optimized processing configuration.

### 3.3. Compression Behavior

The compressive response of the TPMS-sandwich specimens is examined through representative stress–strain curves obtained under quasi-static loading according to ASTM C365. Figure 12a,b report the families of curves for the two extrusion temperatures, while the subsequent discussion focuses on the characteristic regions of the curves and on how layer height modulates stiffness, first-collapse stress, and post-collapse evolution.

All compressive stress–strain curves exhibit a characteristic multi-stage response typical of architected cellular solids:Elastic region: At small strains, the response is linear and governed by the global stiffness of the gyroid core. All configurations show nearly identical initial slopes, indicating comparable elastic moduli.Onset of collapse (first collapse): A well-defined peak marks the buckling or instability of the gyroid walls. This first-collapse stress represents the true strength of the cellular core in compression and is strongly affected by printing parameters.Progressive densification (post-collapse softening or plateau-like region): Immediately after the peak, the stress either slightly drops or remains almost flat as the collapse band spreads through the structure. In this stage, the gyroid walls fold, bend, and progressively close the internal voids. Unlike classical foams, the gyroid does not form a long plateau; instead, it exhibits a short densification region where deformation localizes and the structure becomes increasingly compact.Final compaction (sharp stress rise): Once the voids are almost fully closed and the collapsed walls have packed together, the structure behaves as a quasi-solid. At this point, the stress increases steeply, producing the pronounced upward turn visible in all curves at large strains (ε ≈ 0.45–0.55). This final rise marks the transition from geometrically governed collapse to the intrinsic compressive stiffness of the compacted material.

Overall, the curves reflect the smooth, continuous morphology of the gyroid: failure proceeds through stable folding and densification rather than sudden, brittle crushing, and the transition into the compaction regime occurs earlier than in conventional open-cell lattices due to the absence of thin struts and the presence of continuous curved walls.

At 200 °C, the three stress–strain curves share the same overall shape, but their relative position reveals clear differences in structural stability. The curve corresponding to the smallest layer height consistently lies above the others, indicating a more resistant and stable core during both the collapse and densification stages. As layer height increases, the curves shift progressively downward, signaling less stable wall buckling and a reduced ability to sustain load after the first collapse.

Despite this hierarchy, the general sequence of elastic rise, collapse, short densification, and final compaction remains the same for all configurations; what changes is the height of the curves, not their shape. The finest layer height produces the stiffest and most stable response, whereas the coarsest layer height exhibits an earlier drop in stress and a slightly smoother transition into compaction.

At 220 °C, the curves maintain the same relative ordering observed at 200 °C: the finest layer height again yields the uppermost curve, while the coarsest height produces the lowest one. However, the entire family of curves shifts downward compared to their 200 °C counterparts, revealing a moderate reduction in collapse resistance associated with the higher extrusion temperature.

The softening region following collapse is slightly smoother at 220 °C, suggesting more homogeneous filament fusion and fewer local imperfections in the TPMS walls. Nonetheless, the characteristic sequence of stages remains unchanged, and the distinction between layer heights is still apparent, even though the differences are somewhat less pronounced than at the lower temperature.

Comparing temperatures at a fixed layer height shows a consistent effect: the 220 °C curves lie below the 200 °C ones across the entire deformation range. This downward shift is evident immediately after the elastic region and becomes particularly clear at the onset of collapse. While the qualitative shape of the curves remains identical at the two temperatures, the higher extrusion temperature leads to slightly less stable walls and an earlier transition from collapse to densification.

Importantly, this temperature effect does not modify the ordering imposed by layer height: at both 200 °C and 220 °C, finer layers provide the most stable compressive response, and coarser layers the least. Temperature acts as a vertical offset, uniformly lowering all curves, whereas layer height governs their relative elevation.

The qualitative comparison of the compression curves shows that layer height exerts the dominant influence on the structural stability of the TPMS core. Finer layers generate more uniform walls, with fewer geometric imperfections, and therefore sustain higher stresses during collapse and densification. Coarser layers, by contrast, produce walls with slightly larger waviness and less regular deposition, resulting in lower collapse resistance and a smoother transition into densification. These differences manifest as a clear vertical separation between curves at both temperatures.

Extrusion temperature plays a secondary but systematic role. Increasing the temperature slightly shifts all curves downward, suggesting that higher thermal input smooths or rounds the extruded walls, subtly reducing their ability to resist buckling. Despite this reduction in collapse strength, the overall shape of the curves remains unchanged: the transition from elastic rise to collapse, followed by short densification and final compaction, is consistently preserved for all process conditions. Temperature therefore affects the magnitude of the response but not its mechanism.

Taken together, the compressive behavior reflects the characteristic mode of deformation of gyroid architectures. The absence of discrete struts and the presence of continuous curvature promote stable wall folding and early densification, avoiding catastrophic collapse and ensuring a gradual redistribution of load. The clear influence of layer height, combined with the moderate effect of temperature, indicates that geometric fidelity of the walls, rather than interlayer adhesion, governs compressive performance.

The trends observed in compression complement those previously identified in bending and collectively reveal two distinct process–structure–property regimes within the same TPMS sandwich architecture.

Under bending, the response is skin-dominated. Peak stress and post-peak evolution depend primarily on the continuity, integrity, and bonding of the printed facesheets and on the quality of the skin–core interface. Here, layer height influences the effective cross-section of the outer skins, while extrusion temperature modulates interlayer diffusion and skin consolidation. The result is a complex interaction where the best configuration in bending emerges from a balance between improved skin strength and controlled bonding.

Under compression, by contrast, the response is core-dominated. The load is carried directly by the gyroid walls, and performance depends on their geometric stability and smoothness. Layer height determines the regularity of wall formation, while extrusion temperature acts mostly as a secondary geometric modifier. Interlayer adhesion plays a much smaller role compared to bending, and the ordering of the curves remains the same across temperatures. The collapse mechanism—wall buckling followed by densification—remains stable, predictable, and largely unaffected by skin behavior.

Together, these two regimes highlight the multi-axial nature of process–property relationships in TPMS sandwiches. A printing condition that optimizes bending strength does not necessarily maximize compression resistance, and vice versa. By analyzing both tests, the study provides a broader process window in which temperature and layer height can be tuned according to the functional loading scenario: stiffer skins and strong interfaces for bending, or more uniform walls for compression.

Figure 13 reports the average first-collapse stress for each layer height at the two extrusion temperatures. The qualitative trends observed in the stress–strain curves are fully confirmed here. At both temperatures, the configuration with the finest layer height produces the highest collapse stress, and the values decrease progressively as the layer height increases. This monotonic ordering underlines the central role of vertical discretization in determining the stability of the TPMS walls under axial load.

A second clear trend emerging from the histogram is the effect of extrusion temperature. For every layer height, the bars corresponding to 220 °C lie below those at 200 °C, indicating a uniform reduction in collapse resistance when the extrusion temperature is increased. This downward shift is modest but systematic and mirrors the behavior seen in the curve families: temperature affects the magnitude of the collapse stress but does not alter the ordering imposed by layer height.

The error bars also highlight an important aspect of the compressive behavior. Dispersion remains low and comparable across all configurations, suggesting that the collapse mechanism is highly repeatable and that process-induced variability has a limited influence on the structural stability of the TPMS walls. This is consistent with the smooth, continuous nature of the gyroid geometry, which tends to localize deformation in a predictable manner.

Overall, the histogram reinforces the conclusion that layer height is the dominant parameter governing compression strength, while temperature acts chiefly as a uniform vertical offset, slightly lowering performance but without modifying the underlying mechanical hierarchy established by the geometric discretization.

The residual diagnostics for compressive first-collapse stress are shown in Figure 14. The normal probability plot is nearly linear, with only minor deviations at the extremes; no outliers or influential points are visible. The histogram is approximately symmetric and centered near zero, indicating reasonably normal residual distribution. In the residuals-versus-fitted plot, no curvature or funneling is observed: the spread of the residuals remains constant across the fitted range, suggesting homoscedasticity. The residuals-versus-order panel shows an alternating scatter without drift or cyclic patterns, supporting independence of the observations. Overall, the assumptions required for ANOVA, normality, constant variance, and independence, are satisfactorily met, and no transformation of the data is needed.

The ANOVA table for first-collapse stress confirms that both extrusion temperature and layer height have statistically significant main effects, whereas their interaction is not statistically significant (see Table 7). The linear terms collectively explain the vast majority of the variance, and the model achieves a good fit (R^2^ ≈ 85%), with acceptable adjusted and predicted R^2^ values.

Layer height emerges as the dominant factor, contributing the largest portion of explained variance. This is consistent with the qualitative behavior of the curves, where finer layers consistently provided higher collapse stresses. The significant F-value for layer height reflects the structural sensitivity of the TPMS walls to vertical discretization: smaller layers promote more uniform deposition, reduced geometric irregularities, and more stable buckling behavior.

Temperature also exerts a significant but secondary effect. Its influence is uniform across layer heights: for any given geometry, higher extrusion temperature results in a downward shift of the curve. The significant F-value associated with temperature confirms this systematic reduction in collapse stress, in line with the visual ordering of the histogram. This behavior likely originates from subtle geometric smoothing and reduced dimensional sharpness of the extruded walls at higher thermal input.

In contrast, the interaction term is not statistically significant, indicating that temperature affects all layer heights similarly. This further confirms the trends observed in the stress–strain curves, where temperature acted as a vertical offset rather than modifying the relative ordering established by layer height. The stability and consistency of this response across configurations reinforce the notion that compression is primarily governed by geometric fidelity of the TPMS walls, with interlayer diffusion playing a minimal role.

In summary, the ANOVA reveals a clear, hierarchical influence of process parameters on compressive behavior: layer height is the primary driver of collapse strength, temperature is a secondary but systematic modifier, and no interaction is present, reflecting the inherently core-dominated nature of the compressive response.

The failure morphology observed under compression reflects the progressive and highly stable collapse mechanism characteristic of gyroid TPMS cores. A representative sequence of deformation stages is shown in Figure 15, illustrating the evolution from the undeformed configuration to post-collapse and finally to densification.

In the undeformed state, the gyroid walls maintain their continuous, smoothly curved geometry, with clearly defined passages and minimal filament irregularities. This morphology explains the consistent elastic response observed across all printing conditions: the structure behaves as a homogeneous cellular solid prior to the onset of buckling.

Immediately after first collapse, the deformation localizes within a horizontal band across the specimen. The walls within this band undergo coordinated buckling and bending, accompanied by local rotations and the formation of flattened regions between adjacent surfaces. Importantly, the collapse remains stable and non-catastrophic: no brittle cracking is observed, nor is there any sudden fragmentation of the TPMS walls. Instead, the structure accommodates deformation through progressive folding, which is consistent with the smooth post-collapse behavior recorded in the stress–strain curves.

As loading continues, the specimen enters the densification stage, where the previously buckled walls begin to contact and pack against each other. The gyroid passages become constricted, and the curved surfaces lose their distinct topology as voids close. This progressive impingement of folded walls marks the transition from geometric collapse to solid-like compaction. The deformation becomes more uniform across the height of the sample, and the stress rises sharply once the remaining porosity is exhausted—precisely matching the steep terminal increase in the mechanical curves.

Across all processing conditions, the failure morphology remains qualitatively identical: compression is governed by wall buckling, folding, and packing, rather than by fracture or interfacial separation. This behavior highlights the inherent stability of gyroid architectures under axial loading and explains why temperature and layer height influence primarily the magnitude of collapse stress while leaving the fundamental mechanism unchanged.

## 4. Discussion

The combined analysis of flexural strength, strain at failure, and fracture morphology provides a comprehensive view of how extrusion temperature and layer thickness affect the mechanical behavior of PLA–flax TPMS sandwich structures.

At 200 °C, the maximum stress as layer thickness rises from 0.16 to 0.28 mm, while the strain decreases. The 0.16 mm specimens display early initiation of tensile cracks in the lower skin, followed by partial delamination and upward propagation through the gyroid wall. Limited interlayer fusion at this temperature results in weak bonding between adjacent filaments, promoting premature decohesion and explaining both the lower stress and higher strain at break, which are typical of progressive and non-catastrophic failures. The fracture surfaces confirm an adhesive–cohesive mixed mode, with interfacial separation prevailing near the skin–core interface.

At 0.24 mm, the stress remains comparable, but the delaminated region becomes more extensive, indicating that the lower number of deposited layers does not fully compensate for the reduced interlayer adhesion. The strain values also remain moderate, suggesting that the structure still deforms plastically before failure but with limited energy absorption.

The 0.28 mm configuration at 200 °C shows a more cohesive fracture, localized in the central region with minor secondary cracks. The higher layer thickness favors better thermal retention during deposition, allowing improved filament fusion and a more continuous load transfer across the skin–core interface. This results in the highest stress among the 200 °C samples and in a slightly reduced strain, consistent with stiffer, more integrated behavior.

When the extrusion temperature increases to 220 °C, the flexural strength increases for the 0.16 mm configuration and remains high for 0.24 and 0.28 mm, indicating that elevated temperature generally enhances interlayer bonding without drastically altering the strength ranking. The improvement is most evident at 0.16 mm, where the same fine-layer geometry that penalized bonding at low temperature now benefits from higher heat accumulation, yielding a ~20% increase in stress. The corresponding fracture shows a clean and well-confined crack path with minimal delamination, indicating efficient stress transmission through the skin–core interface.

At 0.24 mm and 0.28 mm, the fracture surfaces exhibit cohesive tearing with limited interfacial damage, confirming the beneficial effect of temperature on adhesion. The 0.28 mm/220 °C specimens display the most compact and integrated crack path, fully consistent with the highest measured strength and stable post-peak response. Despite the slightly lower strain values compared with the 200 °C samples, the deformation remains largely elastic up to failure, suggesting a shift from interfacial to cohesive failure mode as bonding improves.

The trends observed in bending at 220 °C are mirrored only in part by the compressive response. While higher temperature increases bonding and improves skin integrity under bending, it has the opposite effect on collapse strength: all compressive curves shift downward at 220 °C, regardless of layer height. This contrast highlights the dual nature of the sandwich structure: the skins benefit from enhanced molecular diffusion at elevated temperature, whereas the TPMS walls are penalized by subtle geometric smoothing and reduced dimensional definition. Nevertheless, the relative ordering imposed by layer height remains unchanged at both temperatures, reinforcing that compression depends primarily on wall morphology rather than on interlayer adhesion.

Overall, these results demonstrate that both extrusion temperature and layer thickness jointly influence the flexural response of TPMS sandwiches. Higher temperatures enhance bonding at all scales, while increased layer thickness reduces the number of weak interfaces and favors thermal accumulation during deposition. Together, these factors cause a transition from progressive interfacial delamination to localized cohesive rupture, yielding higher strength and structural integrity without compromising deformation stability.

A compressive perspective further clarifies these temperature–layer height interactions. Unlike bending, where failure initiates in the tensile skin and is strongly affected by interlayer bonding, compression is governed by the stability of the gyroid walls themselves. At 200 °C, the finest layer height produces the highest and most stable collapse stress, whereas coarser layers exhibit earlier buckling and lower load-carrying capacity. This behavior reflects the strong sensitivity of wall geometry to vertical discretization: smaller layers generate smoother, more uniform walls, while larger layers accumulate geometric waviness that promotes premature local instability. Importantly, all configurations display the same sequence, elastic rise, collapse, short densification, final compaction, but the height of the curve depends almost entirely on layer height, confirming that compression is a strictly core-dominated mechanism.

Beyond the effects of temperature and layer height, the complex curvature of the Gyroid geometry also affects local stress distribution and the evolution of fracture paths. The layer-by-layer planar deposition inherent to FFF creates regions where printed filaments are either aligned or misaligned with the principal bending stresses. In areas where the local surface orientation of the Gyroid wall matches the filament direction, interlayer bonding is more effective and the fracture tends to be cohesive. Conversely, when the deposition plane is oblique to the applied stress, shear and interfacial decohesion are more likely causing cracks to deflect along the curved interfaces of the TPMS wall. Therefore, the fracture trajectory often follows a tortuous path that reflects the intrinsic geometry of the Gyroid rather than a planar propagation front. These effects become more evident at lower temperatures and larger layer heights, where the reduced diffusion between layers enhances anisotropy, whereas at 220 °C and 0.16 mm layers, the improved wetting and chain mobility lead to smoother, more cohesive fracture surfaces.

Overall, the combined bending–compression analysis shows that the mechanical response of TPMS sandwiches cannot be interpreted through a single dominant mechanism. Under bending, the structure behaves as a skin-dominated laminate, where interlayer diffusion, filament coalescence, and skin–core bonding control peak stress and fracture mode. Under compression, the response switches to a core-dominated regime, governed by wall uniformity, buckling stability, and early densification. Layer height therefore plays a dual role: it modulates skin stiffness in bending and governs wall regularity in compression. Temperature, by contrast, strengthens the skins while slightly weakening the core. These cross-effects define a process window in which the choice of printing parameters must be tailored to the expected loading mode, enabling the targeted optimization of bio-based TPMS sandwich structures.

## 5. Sustainability Assessment

In this section, an energy- and carbon-focused gate-to-gate screening is used to compare the six printing configurations investigated (layer height of 0.16, 0.24, and 0.28 mm × nozzle temperatures of 200 and 220 °C) on a fixed 30% density gyroid sandwich geometry. The functional unit is one conforming specimen; upstream stages (polymer production, packaging, transport) and downstream stages (post-processing, end-of-life) stages are intentionally excluded, as the objective is to rank process parameters within an unchanged design and material set. This approach is consistent with current additive manufacturing sustainability practice, where electricity-to-CO_2_ screening is routinely adopted to benchmark parameter sets before conducting on full cradle-to-grave LCAs. Systematic reviews and methodological studies for polymers and FFF identify electricity as the dominant gate-to-gate contributor and endorse such comparative assessments [51,52].

All prints were executed in batches of three specimens per job for each configuration. Printing time was taken from the slicer and checked against job logs; the measured batch mass Wr was recorded on a scale, while the theoretical mass Wt was obtained from the slicer. To convert time into energy, an average electrical power P ≈ 125 W was assumed for PLA on a K1-class CoreXY printer with a 60 °C bed and no actively heated chamber.

While assuming constant average power simplifies comparison across configurations, it is acknowledged that the 220 °C hotend condition entails a slightly higher instantaneous draw than 200 °C. Power-logging data for similar desktop FFF systems indicate that this increase typically amounts to 5–8 W (<6% of the total machine load), as the heated bed and motion system are the dominant contributors to overall energy use. Over multi-hour prints, the resulting variation in total energy per specimen is below 2%, which is within the experimental uncertainty of the measurement. Therefore, adopting a constant 125 W average does not materially affect the comparative energy or CO_2_ indicators discussed in this section, while ensuring a consistent basis for cross-configuration ranking.

This assumption is based on wattmeter measurements and peer-reviewed analyses: after a brief warm-up transient (typically 350–400 W for the bed and nozzle), the steady-state power draw for PLA printing usually falls in the 90–140 W range, with the heated bed cycling as the main variable. The warm-up lasts only a few minutes during multi-hour jobs (<3% of total time and energy), so a job-level average of around 120–130 W is technically sound for long PLA runs on K1-class machines. A compact, component-based energy model for 3D printers further supports the linear time × power approach used here. A sensitivity check (±15 W on P or ±5% on time) does not change the relative ranking across configurations, as expected for such linear models [53,54,55].

Under these assumptions, batch-level energy is E_job_ = P⋅t. For the three layer height settings (same geometry/material; two nozzle temperatures), the slicer-based times yielded 8 h 45 min → 1.094 kWh at 0.16 mm, 6 h 26 min → 0.804 kWh at 0.24 mm, and 5 h 56 min → 0.742 kWh at 0.28 mm. Because each job contains three parts, energy per specimen is 0.365, 0.268, and 0.247 kWh, respectively. Measured batch masses Wr were approximately 109.8 g (0.16 mm), 106.8–108.2 g (0.24 mm), and 113.4–113.6 g (0.28 mm), systematically 4–6% below slicer predictions Wt (114.6, 113.3, and 118.6 g). This shortfall is consistent with typical FFF voiding, and under-fill reported in controlled power-logging studies, where bed cycling largely governs the steady draw while the moving average remains essentially flat during long prints. Normalizing energy by mass gives energy intensity E/m of about 9.9–10.0 Wh g^−1^ at 0.16 mm, 7.4–7.5 Wh g^−1^ at 0.24 mm, and 6.5 Wh g^−1^ at 0.28 mm, with negligible dependence on nozzle temperature. Consequently, raising layer height from 0.16 to 0.28 mm reduces energy per specimen by ≈32% and energy per gram by ≈34%. These trends are fully aligned with recent experimental, and review works showing that layer height and throughput dominate energy efficiency in FFF, whereas moderate nozzle-temperature adjustments at fixed geometry have second-order effects on job-level electricity [55,56,57].

To connect process with properties, energy was further normalized by the measured flexural strength for each of the six configurations. Using the experimental averages (mean ± SD)—200 °C: 19.3 ± 0.47 MPa (0.16 mm), 19.3 ± 0.25 MPa (0.24 mm), 23.2 ± 0.21 MPa (0.28 mm); 220 °C: 22.2 ± 0.61 MPa (0.16 mm), 19.0 ± 0.29 MPa (0.24 mm), 22.2 ± 0.36 MPa (0.28 mm)—the energy per unit strength (E/σ)_specimen_ becomes 0.0189, 0.0139, and 0.0107 kWh MPa^−1^ at 200 °C and 0.0164, 0.0141, and 0.0111 kWh MPa^−1^ at 220 °C for 0.16, 0.24, and 0.28 mm, respectively. Two implications follow. First, 0.28 mm consistently exhibits the lowest energy per unit property (≈0.011 kWh MPa^−1^), i.e., about 40–70% lower than 0.16 mm depending on temperature; thus, time/throughput becomes the decisive sustainability lever as long as strength does not collapse, which is not observed here.

In this context, “energy per unit property” refers to the ratio between total printing energy consumption and the corresponding mechanical performance, expressed either as flexural strength. This indicator provides an eco-efficiency metric that quantifies the energy required to achieve a given level of structural performance.

Second, at 0.16 mm, the higher strength at 220 °C improves E/σ relative to 200 °C, but the advantage remains insufficient to overturn the global ranking governed by printing time. The dominance of layer height over modest thermal adjustments reflects broader findings in FFF parametric and energy-efficiency literature [56,57].

Electricity was converted into CO_2_ using official Italian grid emission factors. As a base case, an operational-mix factor around 0.27–0.33 kg CO_2_ kWh^−1^ was considered, consistent with the ISPRA 2024 national inventory framework and recent transparency/indicator reports; the mid-range value 0.30 kg CO_2_ kWh^−1^ provides a conservative estimate for multi-hour jobs. Under this factor, batch-level footprints are about 0.33 kg CO_2_ (0.16 mm), 0.24 kg CO_2_ (0.24 mm), and 0.22 kg CO_2_ (0.28 mm), or roughly 110, 80, 73 g CO_2_ per specimen, respectively. Using 0.27 or 0.33 shifts the absolute values but does not change the ranking. The reliance on gate-to-gate electricity as a screening proxy is coherent with AM LCA evidence for polymer printing, where electricity typically dominates process-stage impacts and is therefore a robust indicator for comparing parameter sets on a fixed part and material.

Overall, the screening indicates that (i) Wr is 4–6% below Wt across jobs, consistent with porosity in FFF; (ii) layer height is the primary control on time, energy, and CO_2_ for the present geometry/material, with 0.28 mm achieving the lowest Wh g^−1^ and kWh MPa^−1^; (iii) nozzle temperature mainly tunes strength at a fixed layer height and has a marginal influence on job-level energy; and (iv) the ranking is robust to plausible variations in average power, print time, and grid factors. The analysis thus provides a transparent, literature-based foundation for process optimization that remains methodologically defensible while avoiding the scope creep of a full cradle-to-grave LCA [51,52].

At the same time, still from a gate-to-gate perspective, it is possible to conduct an economic assessment comparing the cost-effectiveness of producing a sample with a specific process configuration to others, while also considering the environmental impact associated with its production.

The aim of this analysis is to identify, among the six configurations studied, the one that represents the best compromise in simultaneously reducing cost and environmental impact. To this end, two indicators are introduced and discussed: eco-efficiency and cost per kg of CO_2_ avoided.

Eco-efficiency is an indicator that measures the value a process or product generates relative to environmental impact it produces [58]. Specifically, in this gate-to-gate study on the 3D printing of TPMS sandwich structures in flax fiber-reinforced PLA, eco-efficiency is interpreted as the ability of the printing process to achieve good mechanical performance, assessed by the flexural strength of the samples produced, with lower energy consumption, lower CO_2_ emissions and reduced production costs per sample. A process is therefore more eco-efficient if it requires less energy and material to achieve certain performance levels, generates lower CO_2_ emissions per unit produced, and reduces unit production costs simultaneously at the same time.

Therefore, the higher the eco-efficiency index, the greater the overall sustainability of the process, since, for the same performance, there is less environmental impact and lower costs [59].

In the literature [59,60,61,62], several formulas are available for calculating eco-efficiency based on data such as process performance, environmental impact and production costs. In this specific case, eco-efficiency was quantified by comparing the flexural strength of the sample to the product of the production cost and the quantity (in kg) of CO_2_ equivalent per sample:(6)$$Eco-efficiency=\frac{Flexural\;strenght\;(MPa)}{Production\;cost\;(€)\times {CO}_{2}\;equivalent\;(kg)}$$

The second indicator used in the analysis is the cost per kg of CO_2_ avoided, which expresses how much money must be spent, or how much is saved, to reduce CO_2_ emissions by 1 kg compared to a reference condition [63]. A positive value for this indicator means that reducing emissions requires additional costs, so more is paid to pollute less; a negative value, on the other hand, indicates that reducing emissions is accompanied by economic savings.

In this study, the cost per kg of CO_2_ avoided was calculated by comparing the change in costs to the change in emissions between two process configurations, according to the following relationship:(7)$$cost\;per\;kg\;of\;{CO}_{2}\;avoided=\frac{C_{2}-C_{1}}{{CO}_{2}\left(1\right)-{CO}_{2}(2)}$$
where C_1_ and C_2_ represent the unit costs (€/sample) of the two configurations, respectively, and CO_2_(1) and CO_2_(2) represent the relative emissions (kg CO_2_ equivalent per sample) [64].

The combined analysis of the two indicators highlights a significant distinction between economic convenience and overall process sustainability. In terms of eco-efficiency, the process with a layer thickness of 0.28 mm is the most advantageous: at 200 °C, it reaches a value of 147.75 MPa/€·kg, higher than the 84.88 MPa/€·kg obtained at 0.16 mm and the 119.26 MPa/€·kg at 0.24 mm. Similarly, at 220 °C, the eco-efficiency of the 0.28 mm process stands at 141.38 MPa/€·kg, compared to 97.63 MPa/€·kg (at 0.16 mm) and 117.41 MPa/€·kg at 0.24 mm. The 0.28 mm process combines good mechanical performance with lower production costs and emissions, demonstrating that, compared to other printing parameters, the 0.28 mm configuration maximizes the ratio between useful performance and environmental impact, making it the most sustainable option.

However, assessing the cost per kg of CO_2_ avoided provides a more detailed picture: a comparison between the 0.16 mm and 0.24 mm processes shows that the latter represents a win-win improvement, as reducing emissions by 1 kg of CO_2_ saves around €2.27, thus achieving a reduction in emissions accompanied by economic savings. Conversely, comparisons between the 0.16 mm and 0.28 mm processes, and between the 0.24 mm and 0.28 mm processes, show that while emissions are reduced, the 0.28 mm process entails an additional cost for each kg of CO_2_ avoided. In particular, producing a 0.28 mm sample compared to a 0.16 mm sample entails an extra cost of approximately €1.07 per kg of CO_2_ avoided, while compared to a 0.24 mm sample, the additional cost rises to approximately €16.69 per kg of CO_2_ avoided. This indicates that in these cases, the reduction in environmental impact is not accompanied by a direct economic advantage.

Overall, these results show that maximum eco-efficiency (0.28 mm) does not necessarily coincide with the best immediate economic benefit, but represents a more favorable balance between energy use, mechanical performance and environmental sustainability. The 0.24 mm process, on the other hand, is an optimal intermediate solution, capable of reducing environmental impact compared to the 0.16 mm process without increasing production costs.

This integrated analysis highlights that even within a gate-to-gate approach, the selection of printing parameters requires joint consideration of energy efficiency, technical performance, and economic sustainability in order to identify the overall optimum point of the process.

From a broader trade-off perspective, absolute mechanical performance and sustainability indicators show a nonlinear correlation. The 0.28 mm configuration maximizes flexural strength while simultaneously minimizing energy consumption and CO_2_ intensity, representing the most eco-efficient condition. However, this mechanical and environmental optimum entails a slightly higher specific cost per avoided CO_2_ unit compared to thinner-layer processes, indicating that improved strength and lower emissions are achieved at marginally higher economic effort. Conversely, the 0.24 mm configuration provides an attractive balance: although mechanically intermediate, it achieves a favorable compromise between environmental benefit and production cost, representing the best overall trade-off between technical performance and sustainability. This analysis confirms that additive manufacturing process optimization should rely on multi-objective criteria encompassing energy, environmental, and mechanical performance rather than single-factor metrics.

## 6. Conclusions

This work quantified how extrusion temperature and layer height influence the flexural response and failure mechanisms of PLA–flax TPMS gyroid sandwich structures fabricated by FFF. Across the six printing conditions, the highest flexural strength (≈23 MPa) was achieved at 0.28 mm/200 °C, while the maximum strain at failure (≈0.06 mm/mm) occurred at 0.16 mm/200 °C, reflecting the trade-off between interlayer bonding and deformation capacity. ANOVA confirmed that both factors and their interaction significantly affected strength and strain, with temperature improving molecular interdiffusion and layer height reducing the number of weak interfaces. Fractography linked these findings to a transition from progressive interfacial delamination at lower temperatures and thinner layers toward localized, cohesive rupture as bonding improved—most evident at 220 °C with thicker layers. A gate-to-gate sustainability assessment indicated that layer height governs time, energy, and CO_2_ for the present geometry, with 0.28 mm minimizing energy per unit property and overall footprint; measured masses were 4–6% below slicer estimates, consistent with FFF porosity. These results provide defensible, TPMS-specific process windows that balance performance and efficiency for bio-based PLA–flax cores; future work should examine moisture conditioning, fiber/matrix surface treatments, facesheet design, and broader LCA boundaries to consolidate industrial applicability. In addition, the present work deliberately focused on a limited yet representative printing window for extrusion temperature and layer height, balancing statistical significance and experimental sustainability; future developments will adopt broader multi-parameter, multi-level DoE schemes to capture higher-order interactions and further refine these process–structure–property correlations. Future investigations will expand the process window by including additional manufacturing parameters, such as nozzle diameter, printing speed, or chamber temperature, and complementary mechanical tests (compression, impact, fatigue, and environmental aging), to further consolidate the predictive capability of the proposed framework.

Future work will also include the evaluation of post-processing strategies aimed at improving interlayer bonding, such as thermal annealing, infrared or ultrasonic treatments, applied within temperature windows that preserve the integrity of the flax fibers. These approaches could represent a viable route to further enhance the structural performance of bio-based TPMS composites fabricated by material extrusion.

Beyond flexural performance, the addition of compressive testing in this study revealed that TPMS sandwiches operate according to two distinctly different mechanical regimes. Under bending, the response is skin-dominated: interlayer diffusion, facesheet integrity, and skin–core bonding govern strength and failure mode. Under compression, the behavior becomes core-dominated, controlled by the geometric regularity and buckling stability of the gyroid walls rather than by interlayer adhesion. As a result, layer height emerges as the primary driver of collapse strength, while extrusion temperature acts mainly as a uniform geometric modifier. This duality demonstrates that optimal printing parameters depend on the expected loading scenario: conditions that maximize flexural strength may not coincide with those that maximize compression resistance. The combined insight strengthens the generality of the proposed process–structure–property framework and supports the development of tailored process windows for bio-based TPMS sandwich structures.

## Figures and Tables

**Figure 1 materials-18-05356-f001:**
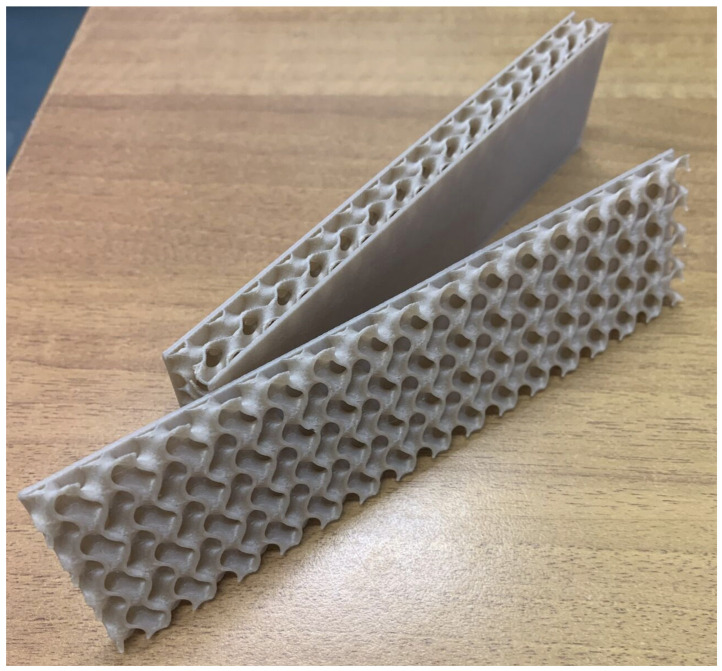
PLA–flax TPMS gyroid sandwich specimens as-printed (FFF, Creality K1 Max): core view and facesheet view. Relative density: 30%; nominal size: 150 × 40 × 12 mm; walls printed at 100% infill; no post-processing.

**Figure 2 materials-18-05356-f002:**
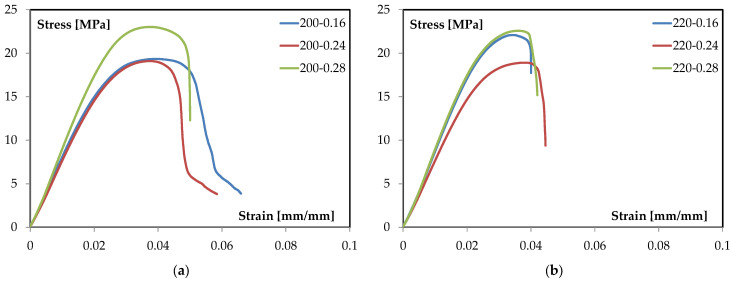
Stress/Strain curves by changing thickness layer for: (**a**) 200 °C and (**b**) 220 °C.

**Figure 3 materials-18-05356-f003:**
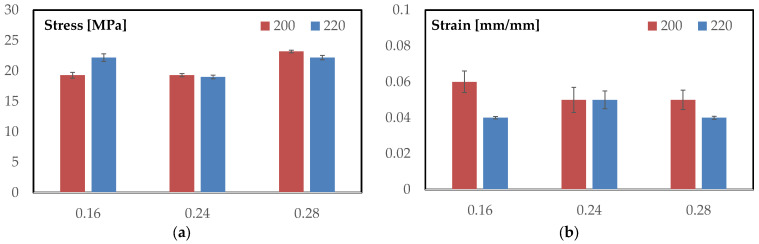
Comparison among: (**a**) maximum stress; (**b**) maximum strain.

**Figure 4 materials-18-05356-f004:**
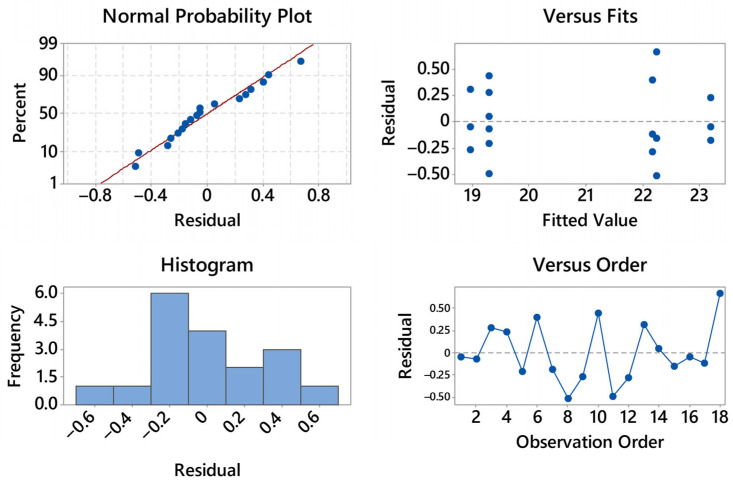
Residual Plots for Stress [MPa].

**Figure 5 materials-18-05356-f005:**
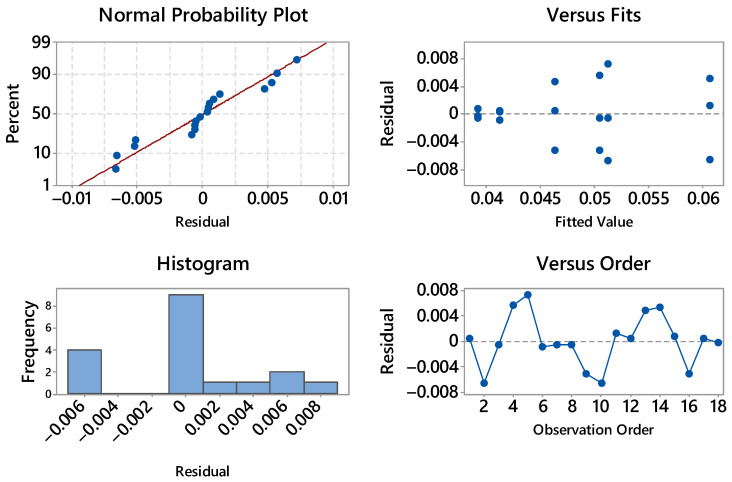
Residual Plots for Strain.

**Figure 6 materials-18-05356-f006:**
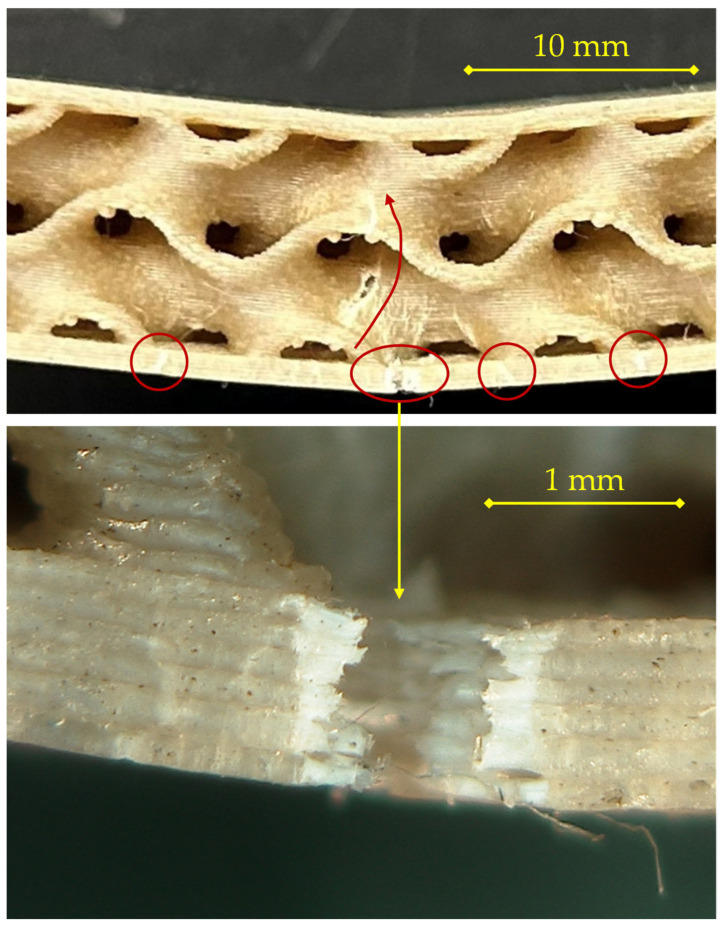
Failure mode for the sample 200|0.16.

**Figure 7 materials-18-05356-f007:**
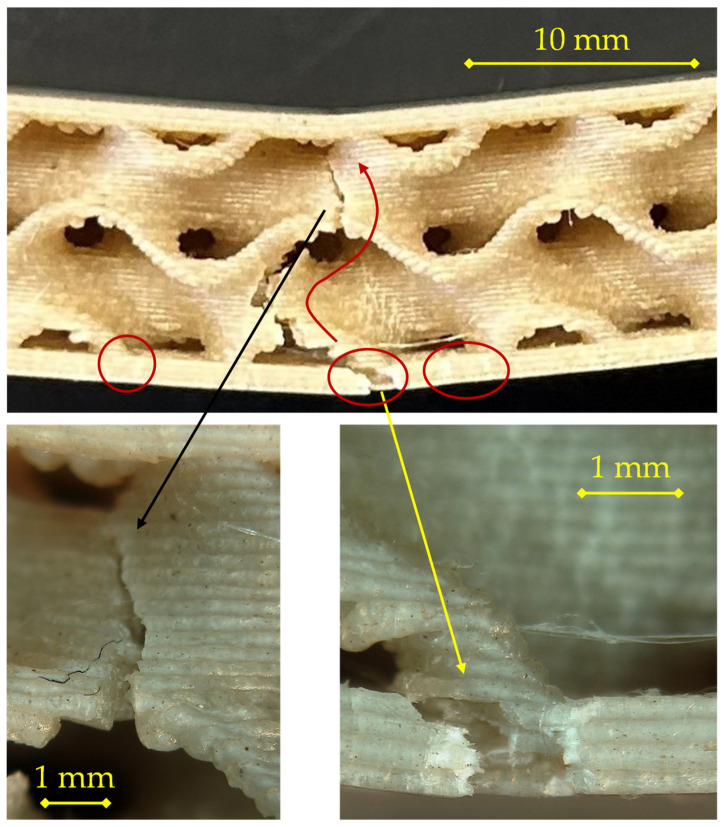
Failure mode for the sample 200|0.24.

**Figure 8 materials-18-05356-f008:**
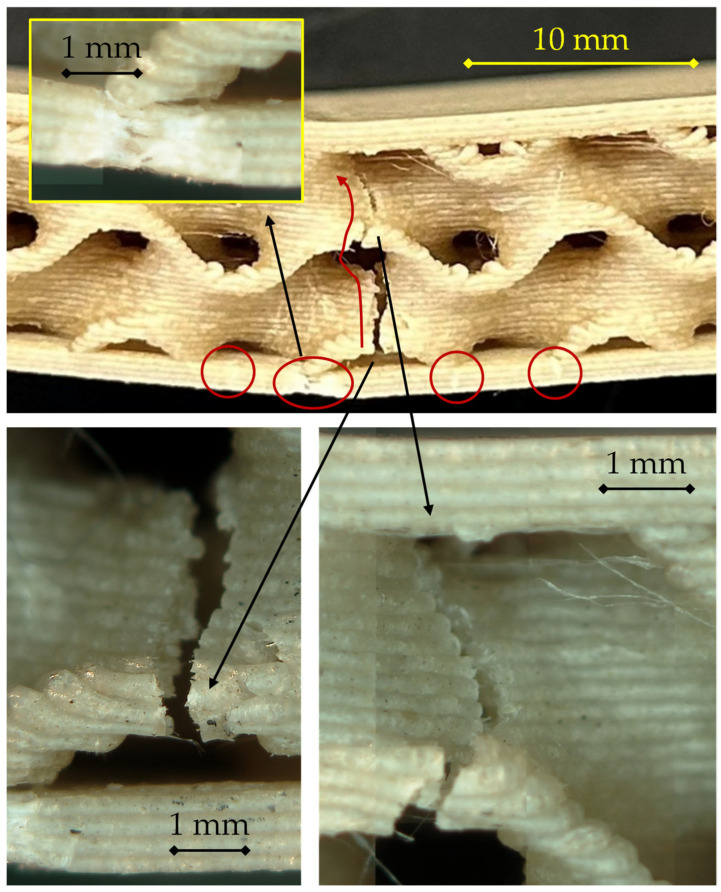
Failure mode for the sample 200|0.28.

**Figure 9 materials-18-05356-f009:**
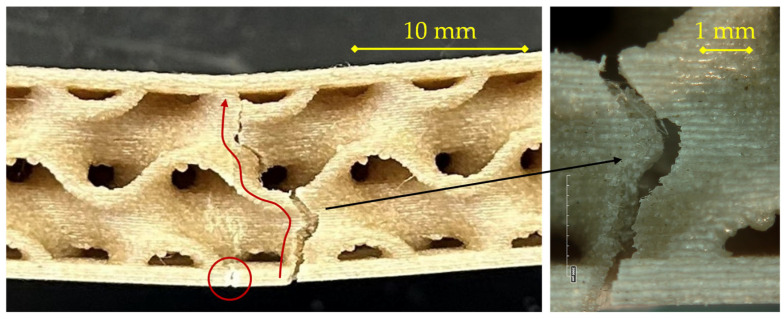
Failure mode for the sample 220|0.16.

**Figure 10 materials-18-05356-f010:**
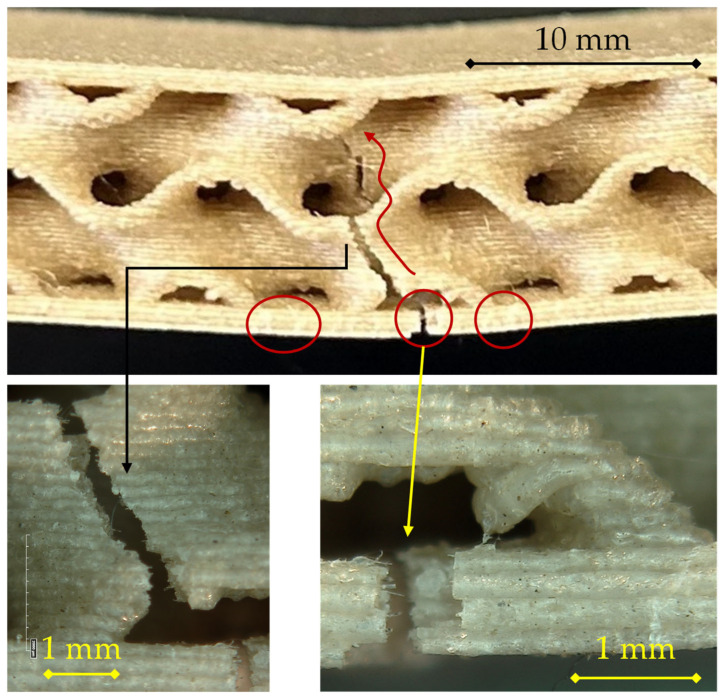
Failure mode for the sample 220|0.24.

**Figure 11 materials-18-05356-f011:**
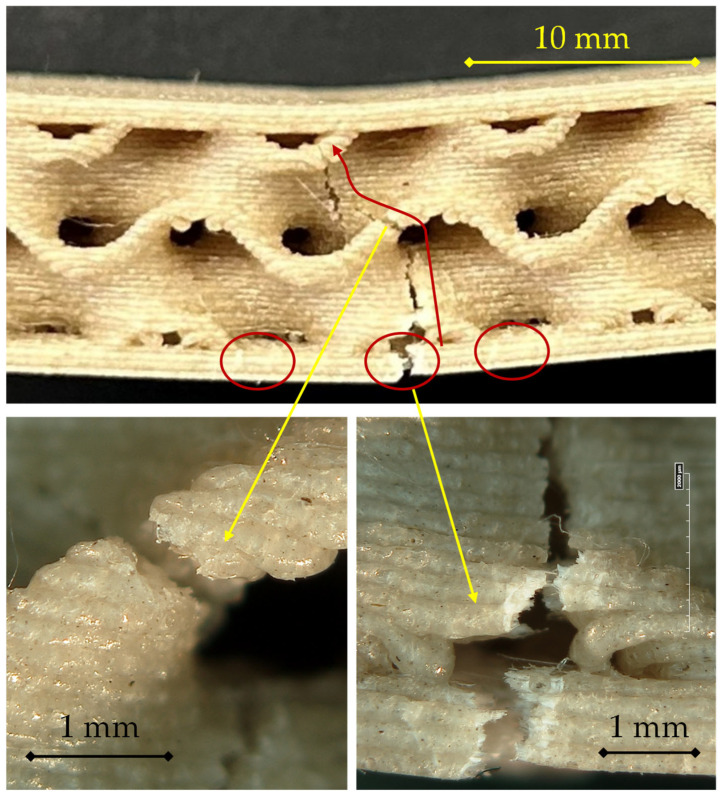
Failure mode for the sample 220|0.28.

**Figure 12 materials-18-05356-f012:**
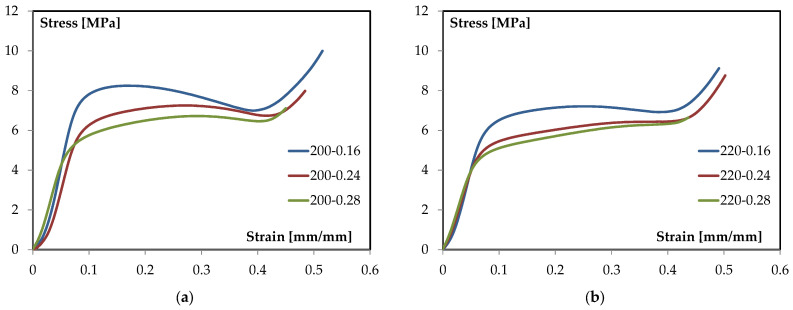
Stress/strain curves by changing thickness layer for (**a**) 200 °C and (**b**) 220 °C.

**Figure 13 materials-18-05356-f013:**
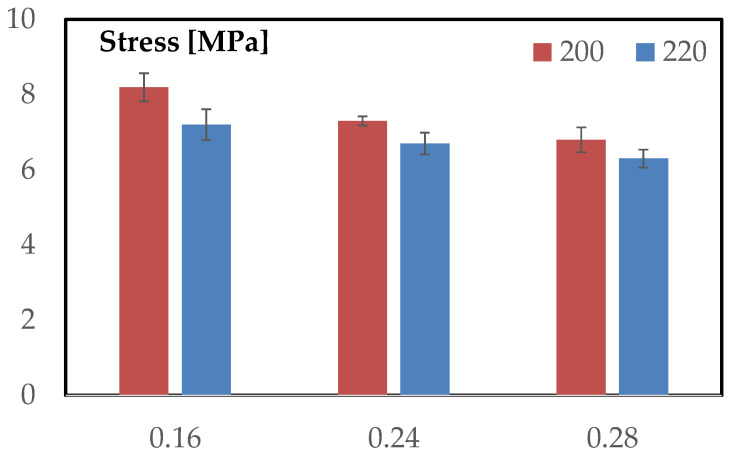
Comparison among first-collapse stresses for all printing configurations.

**Figure 14 materials-18-05356-f014:**
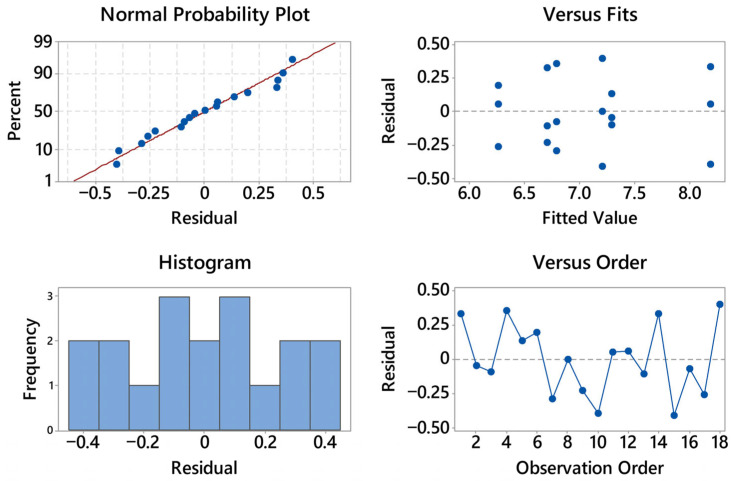
Residual plots for first-collapse stress [MPa].

**Figure 15 materials-18-05356-f015:**
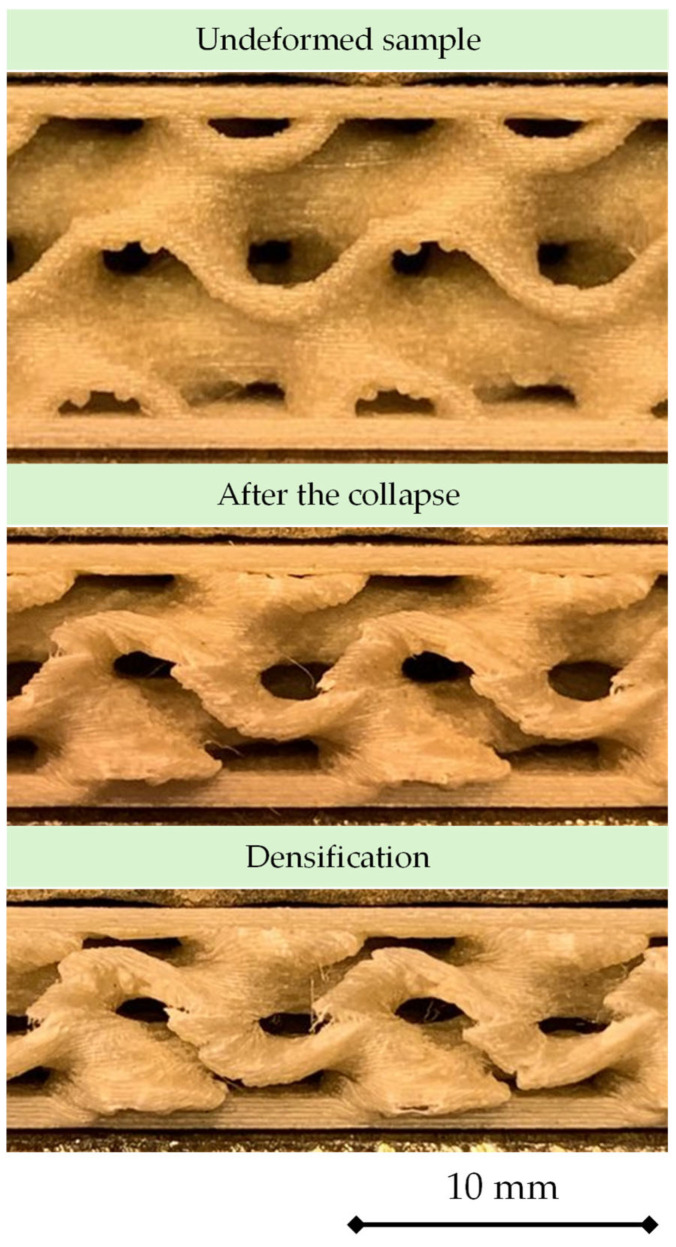
Typical deformation and collapse sequence under compression for a TPMS sandwich (0.16 mm/200 °C configuration).

**Table 1 materials-18-05356-t001:** Printing parameters.

Parameter	Value	Printer
Extrusion Temperature	200|220 °C	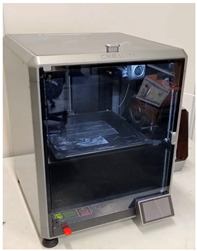
Bed temperature	60 °C	
Layer thickness	0.16|0.24|0.28 mm	
Default line width	0.42 mm	
First layer line width	0.5 mm	
Inner wall line width	0.45 mm	
First layer speed	50 mm/s	
First layer fill speed	105 mm/s	
Outer/Inner wall speed	200/300 mm/s	
Fill speed	270 mm/s	
Resolution	0.012 mm	
Infill patterns	Gyroid (100% wall infill)	

**Table 2 materials-18-05356-t002:** Theoretical printing parameters.

Configuration	Printing Time	Weight (Wt) [g]	Filament Length [m]
200 °C|0.16 mm	8 h 45 min	114.61	38.42
200 °C|0.24 mm	6 h 26 min	113.25	37.97
200 °C|0.28 mm	5 h 56 min	118.64	39.78
220 °C|0.16 mm	8 h 44 m	114.61	38.42
220 °C|0.24 mm	6 h 26 m	113.25	37.97
220 °C|0.28 mm	5 h 56 m	118.64	39.78

**Table 4 materials-18-05356-t004:** Synthesis and comparison.

Configuration	
200 °C|0.16 mm	Lower peak but the most extended post-peak segment (greater deformation capacity after the maximum).
200 °C|0.24 mm	Intermediate behavior; elastic region similar to 0.16 mm, but a shorter tail after the maximum.
200 °C|0.28 mm	Highest peak, shortest tail.
220 °C|0.16 mm	Slightly lower peak than 0.28 mm, similarly short post-peak.
220 °C|0.24 mm	Lower peak and the steepest softening.
220 °C|0.28 mm	Highest peak (≈22–23 MPa), short post-peak.

**Table 5 materials-18-05356-t005:** ANOVA for Stress [MPa].

Source	DF	Adj SS	Adj MS	F-Value	*p*. Value
Model	5	52.841	10.5682	69.38	0.000
Linear	3	39.257	13.0858	85.91	0.000
T [°C]	1	1.255	1.2555	8.24	0.014
h [mm]	2	38.002	19.0010	124.75	0.000
2-Way Interactions	2	13.583	6.7917	44.59	0.000
T [°C] × h [mm]	2	13.583	6.7917	44.59	0.000
Error	12	1.828	0.1523		
Total	17	54.669			

S = 0.390276|R-sq = 96.66%|R-sq(adj) = 95.26%|R-sq(pred) = 92.48%					

**Table 6 materials-18-05356-t006:** ANOVA for Strain.

Source	DF	Adj SS	Adj MS	F-Value	*p*. Value
Model	5	0.000908	0.000182	7.76	0.002
Linear	3	0.000689	0.000230	9.81	0.001
T [°C]	1	0.000636	0.000636	27.18	0.000
h [mm]	2	0.000053	0.000026	1.13	0.355
2-Way Interactions	2	0.000219	0.000110	4.69	0.031
T [°C] × h [mm]	2	0.000219	0.000110	4.69	0.031
Error	12	0.000281	0.000023		
Total	17	0.001189			

S = 0.0048374|R-sq = 76.39%|R-sq(adj) = 66.55%|R-sq(pred) = 46.87%					

**Table 7 materials-18-05356-t007:** ANOVA for first-collapse stress [MPa].

Source	DF	Adj SS	Adj MS	F-Value	*p*. Value
Model	5	6.5904	1.31808	13.97	0.000
Linear	3	6.4045	2.13484	22.63	0.000
T [°C]	1	2.2120	2.21201	23.44	0.000
h [mm]	2	4.1925	2.09625	22.22	0.000
2-Way Interactions	2	0.1859	0.09294	0.98	0.402
T [°C] × h [mm]	2	0.1859	0.09294	0.98	0.402
Error	12	1.1323	0.09436		
Total	17	7.7226			

S = 0.307173|R-sq = 85.34%|R-sq(adj) = 79.23%|R-sq(pred) = 67.01%					

## Data Availability

The original contributions presented in this study are included in the article. Further inquiries can be directed to the corresponding author.

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
