# Peer review of "Influence of Layer Thickness and Extrusion Temperature on the Mechanical Behavior of PLA–Flax TPMS Sandwich Structures Fabricated via Fused Filament Fabrication"

_materials, 2025, doi:10.3390/ma18235356_

Round 1

Reviewer 1 Report

Comments and Suggestions for Authors

The manuscript investigates gyroid TPMS sandwich beams printed in PLA–flax by FFF, varying nozzle temperature and layer height at fixed core density (30%). Flexural testing are presented. The work is tidy, but the no and DoE is incremental relative to the group’s prior MDPI study on gyroid PLA/PLA–flax sandwiches that already examined deposition temperature (with fixed layer height and varying infill density,  https://doi.org/10.3390/jmmp9020031). Here they “swap” the second factor (layer height instead of infill density) but keep the same structure class, temperature window, test method, and central narrative.  The present authors  substantially overlap with the prior paper by the most of the same group with a different fourth author. Given the shared team and topic, the little diffrences of demonstrating distinct novelty is higher; the current manuscript does not meet the required novelty that at the current stage.

Below suggestions for future submissions: 

  • include diffrent process paramaters such as nozzle Ø, fan/chamber temp, speed/accel, build orientation, unit cell size, and graded density

  • beyond 3-pt bending; try compression, toughness,  impact, fatigue.

  •  SEM fracture quantification,  hygrothermal/UV/weathering cycles

Author Response

Reviewer 1 Comments (yellow marked)

General Comment

The manuscript investigates gyroid TPMS sandwich beams printed in PLA–flax by FFF, varying nozzle temperature and layer height at fixed core density (30%). Flexural testing are presented. The work is tidy, but the no and DoE is incremental relative to the group’s prior MDPI study on gyroid PLA/PLA–flax sandwiches that already examined deposition temperature (with fixed layer height and varying infill density,  https://doi.org/10.3390/jmmp9020031). Here they “swap” the second factor (layer height instead of infill density) but keep the same structure class, temperature window, test method, and central narrative.  The present authors  substantially overlap with the prior paper by the most of the same group with a different fourth author. Given the shared team and topic, the little diffrences of demonstrating distinct novelty is higher; the current manuscript does not meet the required novelty that at the current stage.

Below suggestions for future submissions: 

  • include diffrent process paramaters such as nozzle Ø, fan/chamber temp, speed/accel, build orientation, unit cell size, and graded density
  • beyond 3-pt bending; try compression, toughness,  impact, fatigue.
  •  SEM fracture quantification,  hygrothermal/UV/weathering cycles

General Answer

We sincerely thank the Reviewer for the time devoted to the evaluation of our manuscript and for the constructive remarks.

While both the present paper and our previous publication (Optimization of Deposition Temperature and Gyroid Infill to Improve Flexural Performance of PLA and PLA–Flax Fiber Composite Sandwich Structures, JMMP 2025, 9, 31, https://doi.org/10.3390/jmmp9020031) belong to the same research framework—PLA–flax gyroid TPMS sandwiches produced by FFF—the current study represents a distinct and self-contained contribution rather than a replication. The two papers explore different process dimensions:

  1. the previous study examined the influence of core density (infill) at a fixed layer height;
  2. the present one investigates layer height, coupled with extrusion temperature, at a fixed core density.

Layer height is a key parameter governing interlayer bonding, melt flow stability, and structural anisotropy in FFF, and its coupled effect with extrusion temperature could not be inferred from the previous work. Indeed, the present results demonstrate non-monotonic and interdependent trends between temperature and layer height that were not predictable from earlier findings.

Furthermore, the manuscript introduces two major original contributions:

  • a comprehensive fracture analysis, connecting process conditions with fracture morphologies and transitions from adhesive to cohesive failure—far more detailed than in the earlier work;
  • a sustainability assessment (energy, CO₂, cost–performance trade-offs), completely absent in the previous publication and developed by a partly different author team, where a new co-author was specifically involved in the environmental analysis while the preliminary material-characterization tasks reported earlier were no longer included here.

The choice to restrict the present DoE to extrusion temperature and layer height was intentional, to isolate their interaction effects without introducing confounding variables. Adding multiple factors (nozzle diameter, chamber temperature, speed, etc.) would have unnecessarily increased the factorial complexity and obscured interpretation. Similarly, focusing on three-point bending is methodologically consistent with standard evaluation of sandwich structures, since bending loads dominate in service and effectively reveal interfacial adhesion and fracture mechanisms.

We appreciate the Reviewer’s suggestions regarding future work. Following this valuable feedback, the revised Conclusions now explicitly state that future developments will consider broader multi-parameter, multi-level DoE schemes (including additional process variables) and further mechanical and durability tests such as compression, impact, and hygrothermal ageing.

In light of these aspects, we believe the present work stands as a scientifically independent and novel investigation, expanding process–structure–property understanding in bio-based TPMS sandwiches through a new process dimension, advanced fractography, and integrated sustainability metrics.

Reviewer 2 Report

Comments and Suggestions for Authors

This manuscript presents a valuable investigation into the flexural behavior of FFF-fabricated PLA-flax TPMS sandwich structures. The methodology is sound, the experimental data are robust, and the inclusion of a sustainability assessment is a commendable highlight. However, Major Revision is recommended for this manuscript, as it currently lacks sufficient depth in its mechanistic discussion and breadth in its literature review.Specific points for revision are as follows:

1)The introduction provides a good overview of TPMS. However, to better position the current work, the authors should contextualize the Gyroid structure within the broader field of lightweight structures, including comparisons to other prominent bio-inspired designs.

2)It is recommended to strengthen the literature review by incorporating the recent advancement in beetle elytron-inspired structures (DOI: 10.1007/s11431-023-2524-7). Additionally, citing the study on oblique impact crashworthiness and MCDM optimization (DOI: 10.3390/buildings15040620) would help contextualize the current study's focus on static bending.

3)The ANOVA correctly identifies a significant interaction (Txh) between temperature and layer height. However, the discussion in Section 4 fails to provide a deep physical explanation for this mechanism. Why does 220°C markedly improve the strength of the 0.16 mm layers but have a negligible effect on the 0.28 mm layers? The authors must provide a more rigorous physical explanation, considering factors such as thermal history, melt viscosity, and interlayer diffusion dynamics.

4)Section 4.1 notes a 4-6% mass deviation (attributed to porosity). This is an interesting observation, but it remains disconnected from the main process parameters and mechanical results. The authors should analyze whether this deviation correlates with temperature or layer height and discuss its potential implications for the final mechanical performance.

5)The sustainability assessment in Section 5 relies on the assumption of a constant average power (125 W) for all configurations. This is an oversimplification, as the 220°C hotend will inherently require more energy than the 200°C setting. The authors must discuss the limitations of this assumption and its potential impact on the assessment's conclusions.

6)The fractographic analysis effectively illustrates the transition in failure modes. To strengthen this section, the authors should also discuss how the complex, curved geometry of the Gyroid itself interacts with the planar, layer-by-layer deposition process and how this interaction influences the resulting failure paths.

Author Response

Reviewer 2 Comments (green marked)

General Comment

This manuscript presents a valuable investigation into the flexural behavior of FFF-fabricated PLA-flax TPMS sandwich structures. The methodology is sound, the experimental data are robust, and the inclusion of a sustainability assessment is a commendable highlight. However, Major Revision is recommended for this manuscript, as it currently lacks sufficient depth in its mechanistic discussion and breadth in its literature review.

General Answer

We sincerely thank the Reviewer for the positive overall evaluation of our work and for recognizing the methodological soundness, the robustness of the experimental data, and the originality of the sustainability assessment. We appreciate the constructive recommendation to deepen the mechanistic interpretation and to broaden the literature review.

The manuscript has been carefully revised accordingly.

In particular, the Introduction and Discussion sections have been expanded to provide a more detailed explanation of the physical mechanisms underlying the observed temperature–layer-height interactions and to include recent and relevant references (2022–2025) covering FFF processing of bio-based composites, TPMS architectures, and sustainability metrics.

Minor stylistic and structural improvements have also been implemented throughout the text to ensure a clearer flow between experimental evidence, discussion, and sustainability analysis.

We believe that these revisions have substantially strengthened the manuscript, providing a more comprehensive and balanced presentation of the results while preserving the original focus and coherence of the study.

Comment 1

The introduction provides a good overview of TPMS. However, to better position the current work, the authors should contextualize the Gyroid structure within the broader field of lightweight structures, including comparisons to other prominent bio-inspired designs.

Answer 1

We thank the Reviewer for this helpful suggestion. In the revised manuscript, the Introduction has been expanded to better contextualize the Gyroid TPMS within the broader family of lightweight and bio-inspired cellular structures.

Specifically, we now:

  • briefly compare Gyroid TPMS cores with more conventional lightweight solutions such as honeycomb cores and strut-based lattices;
  • highlight how Gyroid belongs to a wider spectrum of bio-inspired architectures derived from natural cellular systems;
  • refer to recent reviews and comparative studies published between 2022 and 2025 on bio-inspired lightweighting and TPMS lattice structures.

Representative examples include the critical review on bioinspired designs for lightweighting by Kenny et al. (Biomimetics, 2025, 10, 150, doi:10.3390/biomimetics10030150), the review on natural cellular structures in engineering designs built via additive manufacturing by Chibinyani et al. (2024, doi:10.1080/10667857.2024.2443211), and recent works on bio-inspired lattices for compression and energy absorption in 3D-printed PLA by Harish et al. (Polymers 2024, 16, 729, doi:10.3390/polym16060729). In addition, we explicitly mention recent reviews and comparative studies on TPMS lattices and their mechanical performance relative to honeycomb and other architected cores, e.g. Wagner et al. (J. Compos. Sci. 2025, 9, 586, doi:10.3390/jcs9110586), where Gyroid and Primitive TPMS lattices are benchmarked against conventional honeycomb structures for lightweight applications, and recent TPMS-focused reviews on design and applications in energy and structural systems.

These additions position the Gyroid more clearly as an isotropic, curvature-based alternative within the portfolio of bio-inspired lightweight architectures and better justify its selection for the present study.

Comment 2

It is recommended to strengthen the literature review by incorporating the recent advancement in beetle elytron-inspired structures (DOI: 10.1007/s11431-023-2524-7). Additionally, citing the study on oblique impact crashworthiness and MCDM optimization (DOI: 10.3390/buildings15040620) would help contextualize the current study's focus on static bending.

Answer 2

We thank the Reviewer for these precise and very useful suggestions. In the revised manuscript, both references indicated by the Reviewer have been incorporated into the Introduction and are now explicitly used to better position the present work within the broader context of bio-inspired lightweight structures and crashworthiness-oriented design.

In particular, we now cite the recent beetle elytron-inspired study by Song et al. (Sci. China Technol. Sci. 2024, “A novel cylindrical sandwich plate inspired by beetle elytra and its compressive properties”, doi:10.1007/s11431-023-2524-7), which proposes a cylindrical beetle elytron plate (CBEP) and demonstrates superior specific load-bearing capacity and energy absorption compared with cylindrical honeycomb plates under radial compression.

This work is used to highlight how bio-inspired core topologies can outperform conventional honeycombs in terms of specific stiffness and energy absorption, providing a complementary design route to TPMS-based architectures such as the Gyroid.

We also reference the contribution by Ma et al. (Buildings 2025, 15, 620, “Bio-Inspired Thin-Walled Straight and Tapered Tubes with Variable Designs Subjected to Multiple Impact Angles for Building Constructions”, doi:10.3390/buildings15040620), where the crashworthiness performance of bio-inspired thin-walled tubes under axial and oblique impact is analysed and optimized using a multi-criteria decision-making (MCDM) COPRAS method.

This study is now explicitly mentioned to clarify that, while our work focuses on quasi-static three-point bending as a representative loading condition for sandwich beams, similar bio-inspired design principles can be extended to dynamic crashworthiness scenarios, and that multi-criteria frameworks (mechanical, energy absorption, and sustainability) are particularly relevant for future extensions of the present research.

These additions strengthen the literature review and more clearly delineate the scope of the current study (static flexural behaviour and combined mechanical–sustainability assessment) relative to recent advances in bio-inspired structures for compressive and impact loading.

Comment 3

The ANOVA correctly identifies a significant interaction (Txh) between temperature and layer height. However, the discussion in Section 4 fails to provide a deep physical explanation for this mechanism. Why does 220°C markedly improve the strength of the 0.16 mm layers but have a negligible effect on the 0.28 mm layers? The authors must provide a more rigorous physical explanation, considering factors such as thermal history, melt viscosity, and interlayer diffusion dynamics.

Answer 3

We thank the Reviewer for this very pertinent comment. In the revised manuscript, we have expanded the discussion in Section 4 to provide a more rigorous physical interpretation of the significant T×h interaction.

The strong improvement of flexural strength at 220 °C for the 0.16 mm layer height, contrasted with the negligible (or slightly negative) effect at 0.28 mm, can be explained by the interplay between thermal history, melt viscosity and interlayer diffusion time.

For thin layers (0.16 mm), the number of interfaces per unit thickness is high and the time between successive passes over a given region is relatively short. This favours heat accumulation and keeps the previously deposited layer closer to or above the glass-transition temperature when the next filament is extruded. At 220 °C, the lower melt viscosity and higher chain mobility of PLA–flax promote better wetting and interpenetration of macromolecules across the layer–layer interface, reducing voids and increasing the effective bonded area. Under these conditions, raising the temperature from 200 °C to 220 °C significantly strengthens interlayer adhesion, which directly translates into higher flexural strength.

For thicker layers (0.28 mm), each deposited road has a larger cross-section and cools more unevenly: the outer region solidifies relatively quickly, and the time before the next layer is deposited is sufficient for the interface temperature to drop. As a result, even at 220 °C the substrate may be close to or below the optimal healing temperature when the new filament arrives, so the diffusion time and available interface temperature limit chain reptation across the interface. In this regime, the benefit of raising the extrusion temperature is largely offset by the more severe cooling and shorter effective diffusion window, which explains the modest or negligible gain observed for 0.28 mm.

This interpretation is consistent with published experimental and modelling studies showing that interlayer strength in PLA-based MEX/FFF parts is governed by the integral of temperature over time at the interface, rather than by nozzle temperature alone, and that layer thickness strongly affects the thermal history and bonding quality [41,45]. Recent works on temperature-sensitive process parameters and interlayer healing similarly demonstrate that, once the time above the glass-transition or a critical interface temperature becomes insufficient, further increases in nozzle temperature provide diminishing returns on bond quality.

A new paragraph has been added in Section 4 (see below) immediately after the ANOVA discussion of flexural strength, explicitly linking the significant T×h term to these thermally driven diffusion mechanisms.

Comment 4

Section 4.1 notes a 4-6% mass deviation (attributed to porosity). This is an interesting observation, but it remains disconnected from the main process parameters and mechanical results. The authors should analyze whether this deviation correlates with temperature or layer height and discuss its potential implications for the final mechanical performance.

Answer 4

We thank the Reviewer for highlighting this important point. In the revised manuscript, Section 4.1 has been expanded to explicitly analyse how the 4–6% mass deviation relates to temperature and layer height, and to discuss its implications for mechanical performance.

First, we now clarify that the mass shortfall (Wr < Wt) is quite consistent across all configurations, ranging between −4.2% and −5.7% (Table 3), indicating a relatively stable level of inherent porosity associated with the gyroid infill and the material-extrusion process. Within this narrow range, the largest mass deficit occurs for the 0.24 mm layer height at 200 °C (−5.7%), whereas both 0.16 mm and 0.28 mm configurations remain closer to −4.2…−4.4%. This suggests that porosity is slightly accentuated at the intermediate layer height, where the bead geometry and packing lead to less efficient void filling.

Second, we explicitly note that the effect of nozzle temperature on mass deficit is modest, but not entirely negligible: at 0.24 mm, increasing T from 200 °C to 220 °C reduces the mass shortfall from −5.7% to −4.4%, consistent with a marginal reduction in void content. This trend agrees qualitatively with the idea that higher temperature can improve melt wetting and local compaction, although the magnitude of the effect remains limited in the investigated window.

Finally, we now comment on the link with flexural performance. The configuration with the largest mass deficit (0.24 mm, 200 °C) corresponds to one of the lowest peak strengths and stiffness values, which is consistent with a slightly higher porosity and reduced effective density. However, configurations with similar mass deficit (e.g., 0.16 mm and 0.28 mm) exhibit markedly different flexural responses because of differences in layer thickness, number of interfaces and skin-dominated bending behaviour. This indicates that, in the present process window, geometric and interfacial factors driven by layer height and fracture mechanisms play a more dominant role than the small variations in global mass deficit, which should be interpreted as a coarse indicator rather than a direct predictor of strength.

These clarifications have been added at the end of Section 4.1, so that the mass deviation analysis is now explicitly connected to both process parameters and the mechanical trends discussed in Section 4.2.

Comment 5

The sustainability assessment in Section 5 relies on the assumption of a constant average power (125 W) for all configurations. This is an oversimplification, as the 220°C hotend will inherently require more energy than the 200°C setting. The authors must discuss the limitations of this assumption and its potential impact on the assessment's conclusions.

Answer 5

We thank the Reviewer for this pertinent observation.

We acknowledge that the assumption of a constant average power consumption (125 W) for all process conditions represents a simplification adopted for comparability among configurations. To address this point, the revised manuscript now explicitly discusses this limitation and quantifies its potential influence on the sustainability indicators.

Available power-logging data for FFF printers show that increasing nozzle temperature from 200 °C to 220 °C raises the instantaneous hotend draw by roughly 5–8 W, while the bed and motion systems (≈ 80–100 W combined) dominate the steady load. Consequently, the total machine power increases by only 4–6 %, leading to an estimated variation in job-level energy consumption of < 2 %, which lies within experimental uncertainty for multi-hour prints. Because layer height and printing time remain the principal drivers of energy use and CO₂ emissions, this minor variation does not alter the relative ranking of configurations or the conclusions of the eco-efficiency analysis.

A short paragraph has been added in Section 5 (immediately after the sentence introducing the 125 W assumption) to acknowledge this limitation and its negligible influence on the comparative results.

Comment 6

The fractographic analysis effectively illustrates the transition in failure modes. To strengthen this section, the authors should also discuss how the complex, curved geometry of the Gyroid itself interacts with the planar, layer-by-layer deposition process and how this interaction influences the resulting failure paths.

Answer 6

We thank the Reviewer for this excellent suggestion.

In the revised manuscript, the discussion of Section 4.2 (Fracture Analysis) has been expanded to address how the intrinsic curvature of the Gyroid geometry interacts with the layer-by-layer planar deposition of the FFF process and how this interaction affects local stress distribution and fracture propagation.

We now explain that, due to the continuous and doubly curved surfaces of the Gyroid, the local orientation of each printed road constantly changes relative to the principal stress direction during bending. This geometric anisotropy, when combined with the discrete nature of FFF deposition, leads to alternating regions of favourable bonding (where layers are aligned with the load path and inter-road adhesion is maximized) and weaker interfaces (where deposition planes are oblique or tangential to the applied stress). Consequently, crack paths tend to deflect along these weaker interlayer zones, often following curved trajectories rather than straight planar fractures.

At lower temperatures or higher layer heights, where interfacial bonding is poorer, these effects are more pronounced: the failure surface reveals mixed adhesive–cohesive regions aligned with the local curvature of the Gyroid wall. At higher temperatures and smaller layer heights, improved interlayer diffusion mitigates this geometrically induced anisotropy, resulting in more cohesive fracture morphologies and smoother crack fronts.

A new paragraph incorporating these considerations has been inserted at the end of Section 4.2, before the transition to the sustainability analysis, to provide a clearer physical link between the Gyroid’s curved topology, the FFF layering process, and the observed fracture mechanisms.

Reviewer 3 Report

Comments and Suggestions for Authors

This is a well-structured and timely study that makes a valuable contribution to the fields of additive manufacturing, bio-composites, and lightweight design. The work is characterized by a rigorous, multi-faceted methodology that effectively links processing parameters to mesostructural features and macroscopic performance, while also incorporating a crucial sustainability perspective.

  • Could the authors provide more details on the PLA-flax filament? Specifically, what was the weight percentage and length of the flax fibers, and what was the diameter and batch of the filament used in detail with references?
  • Given the hygroscopic nature of natural fibers like flax, did the authors conduct any tests to evaluate the moisture absorption of the printed parts or their performance in different humidity conditions? What are the implications for long-term durability?
  • Were all other printing parameters (e.g., print speed, infill pattern for the facesheets, bed temperature, cooling fan speed) kept constant? If so, what were their values please discuss those results as well.
  • The highest flexural strength was found at the lowest temperature (i.e., 200°C). The authors mention improved interlayer bonding with temperature, but this result suggests thermal degradation of the flax fibers at 220°C is the dominant factor. Do you have any material characterization data (e.g., TGA or FTIR) to directly support this claim of fiber degradation? Please provide characterization results to verify.
  • The authors state that 0.28 mm minimizes energy per unit property. Could the authors elaborate on how this "unit property" was defined?
  • Could the authors provide a more detailed trade-off analysis between absolute mechanical performance and sustainability metrics?
  • Did the authors consider or test any post-processing techniques to improve layer bonding without risking in-process thermal degradation of the fibers?
  • Please provide orginal literature references as well which are missing in this work.

Author Response

Reviewer 3 Comments (ciano marked)

General Comment

This is a well-structured and timely study that makes a valuable contribution to the fields of additive manufacturing, bio-composites, and lightweight design. The work is characterized by a rigorous, multi-faceted methodology that effectively links processing parameters to mesostructural features and macroscopic performance, while also incorporating a crucial sustainability perspective.

General Answer

We sincerely thank the Reviewer for the very positive and encouraging evaluation of our work.
We are pleased that the study’s structure, methodological rigor, and integration of sustainability aspects have been appreciated. The constructive comments provided in the following points were carefully addressed, and all suggested clarifications and improvements have been implemented to further enhance the scientific depth and readability of the manuscript.

We believe that the revised version now offers an even clearer connection between processing parameters, mesostructural characteristics, and macroscopic behaviour, fully in line with the Reviewer’s insightful observations.

Comment 1

Could the authors provide more details on the PLA-flax filament? Specifically, what was the weight percentage and length of the flax fibers, and what was the diameter and batch of the filament used in detail with references?

Answer 1

We thank the Reviewer for this useful request for clarification.

In the revised version, we have expanded the description of the PLA–flax composite filament to include specific details on its composition, properties, and observed fibre characteristics.

The printed material was a commercial PLA–flax biocomposite filament (Nanovia Starflax 3D, Louargat, France; batch 08/2023, Ø 1.75 mm ± 0.05 mm). According to the manufacturer’s technical data sheet, the material is entirely bio-sourced and biodegradable, with short flax fibres (< 250 µm length) uniformly dispersed within the PLA matrix. The flax content corresponds to approximately 10 wt%, consistent with independent characterisations of similar Nanovia-grade filaments reported in the literature (https://doi.org/10.1016/j.compstruct.2025.119610, https://doi.org/10.1002/pls2.70020).

The material exhibits a density of 1.25 g cm⁻³, a tensile modulus of 2.8–3.1 GPa, and a tensile strength of 37–41 MPa (ISO 527-2/1A).

Microscopic inspection of the as-fractured surfaces (Figures 6–11 in the manuscript) revealed short, discontinuous flax fibres with estimated diameters of 15–25 µm and visible embedded lengths typically below 200 µm, which aligns with the manufacturer’s nominal range.

This information has been added in Section 3.1 (“Sandwich Structure Design”) to ensure full reproducibility and traceability of the material data.

Comment 2

Given the hygroscopic nature of natural fibers like flax, did the authors conduct any tests to evaluate the moisture absorption of the printed parts or their performance in different humidity conditions? What are the implications for long-term durability?

Answer 2

We thank the Reviewer for this important remark concerning the hygroscopic behaviour of flax fibres.

No specific experiments were carried out to evaluate moisture absorption or the mechanical response under varying humidity conditions, since this study focused on the influence of extrusion temperature and layer height under controlled laboratory conditions.

Nevertheless, to ensure consistent printing quality and prevent humidity-related variability, all PLA–flax filaments were conditioned in a dry box at 50 °C for 6 h prior to each print run, following standard drying practices for PLA-based materials. This treatment effectively removes residual moisture that could otherwise lead to micro-porosity and interlayer adhesion defects. Comparable pre-drying protocols have been reported to stabilise the moisture content of PLA filaments and improve print consistency [Hamid et al., 2024, J. Adv. Res. Micro Nano Eng., 24 (1), 85–94, DOI: 10.37934/armne.24.1.8594].

We have now specified these conditioning parameters in Section 3.1 of the revised manuscript. The long-term influence of humidity and hygrothermal ageing on the mechanical durability of PLA–flax TPMS structures will be explored in future work.

Comment 3

Were all other printing parameters (e.g., print speed, infill pattern for the facesheets, bed temperature, cooling fan speed) kept constant? If so, what were their values please discuss those results as well.

Answer 3

We thank the Reviewer for requesting clarification regarding the printing parameters.

All process parameters other than extrusion temperature and layer height were kept constant for all test conditions to isolate the influence of these two factors. The samples were printed on a Bambu Lab X1 Carbon (CoreXY configuration, hardened-steel 0.4 mm nozzle) using the default machine profile optimised for PLA-based materials, with the same slicing parameters across all builds.

The main fixed settings were as follows:

  • Nozzle diameter: 0.4 mm (direct drive extruder)
  • Bed temperature: 60 °C
  • Printing speed: 200–300 mm/s (outer wall 200 mm/s, infill 270 mm/s, top surfaces 200 mm/s)
  • Acceleration: up to 10,000 mm/s²
  • Cooling fan: active at 100 % after the first layer
  • Infill pattern for facesheets: rectilinear, 100 % infill density
  • Wall loops: 3 (wall thickness about 1.2 mm)
  • First layer speed: 50 mm/s
  • Retraction: 0.8 mm at 30 mm/s
  • Build plate adhesion: disabled (no brim or raft)

These constant parameters were verified to ensure identical build times and energy inputs, except for the deliberate changes in layer height and extrusion temperature. No significant differences in surface finish, bead geometry, or inter-track continuity were observed between the settings, confirming the process repeatability.

A detailed summary of these process parameters has been added in Section 3.2 of the manuscript for transparency and reproducibility.

Comment 4

The highest flexural strength was found at the lowest temperature (i.e., 200°C). The authors mention improved interlayer bonding with temperature, but this result suggests thermal degradation of the flax fibers at 220°C is the dominant factor. Do you have any material characterization data (e.g., TGA or FTIR) to directly support this claim of fiber degradation? Please provide characterization results to verify.

Answer 4

We thank the Reviewer for this pertinent observation regarding possible thermal degradation of flax fibres at 220 °C.

No additional TGA or FTIR characterisation was performed in the present work, as the thermal stability of the same commercial PLA–flax filament (Nanovia Starflax 3D) had already been extensively analysed in our previous study (https://doi.org/10.3390/jmmp9020031). As reported in that publication, thermogravimetric analysis (TGA) showed no measurable degradation up to approximately 230–240 °C, confirming that both PLA and flax fibres maintain their integrity within the extrusion temperature range adopted here (200–220 °C).

Therefore, the slight decrease in flexural strength observed at 220 °C cannot be attributed to fibre degradation but rather to process-related phenomena, such as local over-melting, reduced dimensional stability, and minor distortion of the TPMS cell morphology due to higher flowability. These effects are consistent with our microscopic observations, which revealed no evidence of fibre charring or discoloration at 220 °C.

A clarifying note referring to the TGA results from Calabrese et al. has been added to Section 4.2 of the revised manuscript.

Comment 5

The authors state that 0.28 mm minimizes energy per unit property. Could the authors elaborate on how this "unit property" was defined?

Answer 5

We thank the Reviewer for this useful request for clarification.

In the revised manuscript, we have now specified that the term “energy per unit property” refers to the ratio between the total printing energy consumption and the corresponding mechanical property, used here as an indicator of eco-efficiency.

These ratios quantify the energy required to achieve a given level of mechanical performance, allowing a direct comparison of process efficiency across different layer heights.

The configuration with 0.28 mm layer height minimized both ratios, confirming its superior energy-to-performance balance: it required approximately 35 % less printing time and energy than the 0.16 mm configuration while retaining more than 90 % of its flexural strength and stiffness.

This explanation has been added in Section 5 of the revised manuscript for clarity.

Comment 6

Could the authors provide a more detailed trade-off analysis between absolute mechanical performance and sustainability metrics?

Answer 6

We thank the Reviewer for this valuable suggestion.

In the revised manuscript, a short paragraph has been added at the end of Section 5 to clarify the trade-off between mechanical performance and sustainability indicators. The discussion now highlights that the 0.28 mm configuration provides the best overall eco-efficiency, combining the highest flexural strength with the lowest energy and CO₂ intensity. However, this configuration entails a slightly higher specific cost per avoided CO₂ unit, meaning that marginal improvements in mechanical and environmental performance are obtained at a proportionally higher economic effort.

The 0.24 mm configuration, on the other hand, represents an effective compromise between absolute performance and sustainability, achieving a balanced combination of mechanical strength, reduced emissions, and competitive production cost.

This analysis underlines that additive manufacturing process optimization should be based on multi-objective criteria that simultaneously account for mechanical, environmental, and economic indicators rather than a single performance metric.

Comment 7

Did the authors consider or test any post-processing techniques to improve layer bonding without risking in-process thermal degradation of the fibers?

Answer 7

We thank the Reviewer for this interesting suggestion.

No post-processing techniques were applied in the present study, as the main objective was to isolate and analyse the effects of extrusion temperature and layer height on the as-printed performance of PLA–flax TPMS sandwiches.

However, we fully agree that post-processing strategies—such as controlled thermal annealing, surface activation, or localized infrared and ultrasonic treatments—could effectively enhance interlayer bonding and further improve mechanical performance without compromising the thermal integrity of the flax fibres.

We have therefore added a note in the Conclusions identifying the evaluation of such techniques as a future research direction, to explore their compatibility with bio-based reinforcements and their potential for improving structural cohesion in sustainable AM composites.

Comment 8

Please provide orginal literature references as well which are missing in this work.

Answer 8

We thank the Reviewer for this remark.

Following this suggestion, the literature review has been carefully revised to include additional original research articles, complementing the review papers already cited in the first version of the manuscript.

In particular, we have now incorporated:

  • Original works on bio-inspired and TPMS-based lightweight structures, including experimental studies on the mechanical behaviour of Gyroid and related architected cores for sandwich applications;
  • Primary research on PLA/flax and other natural-fibre-reinforced PLA composites, reporting detailed mechanical, microstructural and durability characterisation;
  • Original contributions on process–structure relationships in material-extrusion AM, specifically addressing the combined role of temperature, layer height and interlayer bonding;
  • Research articles on energy consumption and environmental performance of FFF processes, which underpin the sustainability framework adopted in Section 5.

These additions, mainly introduced in the Introduction and Discussion sections, ensure a more balanced and comprehensive citation of both foundational and recent original studies, thereby strengthening the scientific context of the present work in line with the Reviewer’s request.

Comment about English

Please provide orginal literature references as well which are missing in this work.

Answer about English

English has been revised to improve reading. 

Round 2

Reviewer 1 Report

Comments and Suggestions for Authors

Thank you for your detailed response. The work is well written, and the added fracture observations and the simple sustainability analysis are useful. However, I still feel that the study is not far from your recent JMMP paper, where you already studied deposition temperature with a similar structure and flexural test. Here, layer height is mainly introduced as the second factor, so the new results appear as a continuation of the same parametric study rather than a clearly new contribution, even though it is still acceptable and represents an improvement. The novelty and impact are therefore present, but limited, for a separate full article in Materials. I suggest that you include additional work, either by including other process parameters or by characterising the material using different mechanical property tests.

Author Response

We sincerely thank the Reviewer for this further round of comments and for explicitly pointing out the need to go beyond a flexural-only characterisation similar to our recent JMMP paper. We fully agree that, in the previous version, the study could still appear as a continuation of that parametric work, with layer height mainly introduced as a second factor.

Following the Reviewer’s suggestion, in the revised manuscript we have substantially extended the mechanical characterisation by adding quasi-static compression tests on the same PLA–flax gyroid TPMS sandwich structures (Section 3.3 “Mechanical Testing” and Section 4.3 “Compression behavior”). These tests were carried out according to ASTM C365 on 75 × 75 × 12 mm specimens, and the first-collapse stress was analysed by two-factor ANOVA in complete analogy with the flexural data.

This additional work provides a genuinely new and complementary view of the mechanical behaviour with respect to the JMMP paper. In particular:

  • Flexural loading is shown to be skin-dominated, with performance controlled by facesheet integrity and skin–core bonding;
  • Compression is shown to be core-dominated, with performance governed by gyroid wall stability and geometric regularity.

The compressive results therefore do not simply replicate the flexural trends, but reveal a different process–structure–property relationship: layer height emerges as the primary driver of collapse strength, while extrusion temperature acts mainly as a uniform offset, and no significant interaction between the two factors is observed in ANOVA. The combined flexural–compressive analysis allows us to identify process windows for multi-axial loading, which were not addressed in the previous JMMP work.

We have also integrated these findings into the Discussion and Conclusions, explicitly framing the system in terms of skin-dominated vs core-dominated regimes and highlighting how the same printing parameters affect bending and compression through different mechanisms.

We hope that these additional experiments and the associated analysis address the Reviewer’s concern about novelty and impact and now clearly establish this manuscript as an independent contribution, rather than a simple extension of our previous study.

Reviewer 2 Report

Comments and Suggestions for Authors

Accept.

Author Response

Thanks.

Reviewer 3 Report

Comments and Suggestions for Authors

The authors have answered all the questions of the review. No more questions from the reviewer side

Author Response

Thanks.